# Beyond Spatio-Temporal Representations: Evolving Fourier Transform for Temporal Graphs

**Anson Bastos**[1,2]**, Kuldeep Singh**[4]**, Abhishek Nadgeri**[3]**, Manish Singh**[2]**, Toyotaro Suzumura**[5]
[1]HERE Technologies, India [2]Indian Institute of Technology Hyderabad, India
[3]RWTH Aachen, Germany [4]Cerence Gmbh, Germany [5]The University of Tokyo, Japan
`ansonbastos@gmail.com, kuldeep.singh1@cerence.com, abhishek.nadgeri@rwth-aachen.de`
`msingh@cse.iith.ac.in, suzumura@acm.org`

## Abstract

We present the Evolving Graph Fourier Transform (*EFT*), the first invertible spectral transform that captures evolving representations on temporal graphs. We motivate our work by the inadequacy of existing methods for capturing the evolving graph spectra, which are also computationally expensive due to the temporal aspect along with the graph vertex domain. We view the problem as an optimization over the Laplacian of the continuous time dynamic graph. Additionally, we propose pseudo-spectrum relaxations that decompose the transformation process, making it highly computationally efficient. The *EFT* method adeptly captures the evolving graph's structural and positional properties, making it effective for downstream tasks on evolving graphs. Hence, as a reference implementation, we develop a simple neural model induced with *EFT* for capturing evolving graph spectra. We empirically validate our theoretical findings on a number of large-scale and standard temporal graph benchmarks and demonstrate that our model achieves state-of-the-art performance.

## 1 Introduction

In numerous practical situations, graphs exhibit temporal characteristics, as seen in applications like social networks, citation graphs, and bank transactions, among others (Kazemi et al., 2020). These temporal graphs can be divided into two types: 1) temporal graphs with constant graph structure (Grassi et al., 2017; Cao et al., 2020), and 2) temporal graphs with dynamic structures (Zhou et al., 2022; Bastos et al., 2023; da Xu et al., 2020). Our focus in this work is the latter case.

The evolving graphs have been comprehensively studied from the spatio-temporal graph-neural network (GNN) perspective, focusing on propagating local information (Pareja et al., 2020; Shi et al., 2021; Xiang et al., 2022; da Xu et al., 2020). Albeit the success of spectral GNNs for static graphs for capturing non-local dependencies in graph signals (Wang & Zhang, 2022), they have not been applied to temporal graphs with evolving structure. To make spectral GNN work for temporal graphs effectively and efficiently, there is a necessity for an invertible transform that collectively captures evolving spectra along the graph vertex and time domain. To the best of our knowledge, there exists no such transform in the spectral domain for temporal graphs with evolving structures.

In the present literature, Graph Fourier Transform (GFT), which is a generalization of Fourier Transform, exists for static graphs but can not capture spectra of evolving graph structure (Shuman et al., 2013). Hence, it cannot be applied to temporal graphs due to the additional temporal aspect. One naive extension would be to treat the time direction as a temporal edge, construct a directed graph with newly added nodes at each timestep, and find the Eigenvalue Decomposition (EVD) of the joint graph. However, this would lose the distinction between variation along temporal and vertex domains. Moreover, such an approach would incur an added computational cost by a multiplicative factor of $\mathcal{O}(T^3)$, which would be prohibitively high for the temporal setting with a large number of timesteps. Thus, in this paper, we attempt to find an approximation to the dynamic graph transform that would capture its evolving spectra and be efficient to compute.

We aim to propose a novel transform for a temporal graph to its frequency domain. For this we consider the Laplacian of the dynamic graph and find the orthogonal basis of maximum variation to obtain the spectral transform (Hammond et al., 2011). We view this as an optimization of the variational form of the Laplacian such that the optimal value is within the $\epsilon-$ pseudospectrum (Tao, 2008). We then show that such optimization gives us a simple and efficient to compute solution while also being close to the exact solution of the variational form under certain conditions of Lipschitz continuous dynamic graphs. Effectively, we propose a method to simultaneously perform spectral transform along both the time and vertex dimensions of a dynamic graph. This solves the following challenges over the natural extension of EVD over dynamic graphs: 1) The proposed transformation is computationally efficient compared to the direct eigendecomposition of the joint Laplacian. 2) Distinction between time and vertex domain frequency components with the proposed transform provides interpretability to the transformed spectral domain. We term the proposed concept as "Evolving Graph Fourier Transform" (*EFT* ).

In summary, we make the following key contributions:

- We propose *EFT* (grounded on theoretical foundations), that transforms a temporal graph to its frequency domain for capturing evolving spectra.

- We provide the theoretical bounds of the difference between *EFT* and the exact solution to the variational form and analyze its properties.

- As a reference implementation, we develop a simple neural model induced with the proposed transform to process and filter the signals on the dynamic graphs for downstream tasks. We perform extensive experimentation on large-scale and standard datasets for dynamic graphs to show that our method can effectively filter out the noise signals and enhance task performance against baselines.

## 2   RELATED WORK

**Spectral Graph Transforms:** Work by (Hammond et al., 2011) was among the first to propose a computationally efficient algorithm to compute the Fourier Transform for static graphs. Loukas et al. (Loukas & Foucard, 2016) conceptualized Joint Fourier Transform (JFT) over graphs on which the signals change with time. JFT has been generalized in (Kartal et al., 2022) by proposing the Joint Fractional Fourier Transform (JFRT). However, JFT and JFRT does not consider graph structures evolving with time. (Cao et al., 2021) apply JFT and propose a model for time series forecasting. (Villafañe-Delgado & Aviyente, 2017) summarized graphs over time by using Tucker decomposition to the dynamic graph Laplacian in order to obtain an orthogonal matrix and further applies it to a cognitive control experiment. However, this method does not fully capture the varying graph information in a lossless sense. Researchers have also proposed spectral methods for spatio-temporal applications such as action recognition (Yan et al., 2018; Pan et al., 2020), traffic forecasting (Yu et al., 2017) etc. Other works such as (Mahyari & Aviyente, 2014; Chen et al., 2022; Sarkar et al., 2012; Kurokawa et al., 2017; Jiang et al., 2021; Cheng et al., 2023) also consider temporal graphs, but ignore the evolving structure. We position our work as the novel spectral graph transform for temporal graphs which is currently a gap in existing literature.

**Temporal Graph Representation Learning:** Since static graph methods do not work well with dynamic graphs (Pareja et al., 2020), researchers have proposed a slew of methods (Pareja et al., 2020; Goyal et al., 2020; Xiang et al., 2022), for learning on dynamic graphs for problems such as link prediction and node classification. One elementary way to adapt methods developed for static graphs on dynamic graphs is to use RNN modules in conjunction with GNN modules to capture the evolving graph dynamics. Researchers (Seo et al., 2016; Narayan & Roe, 2018; Manessi et al., 2020) have explored this idea extensively. Some other recent approaches model several real world phenomena, however, these methods rely on an RNN for encoding temporal information such as Bastas et al. (2019), da Xu et al. (2020), Ma et al. (2020), etc. Most generic among these works is TGN (Temporal Graph Networks) (Rossi et al., 2020) that remembers nodes and connections it has seen in the past, and then uses that memory to update new nodes and connections that it hasn't seen before. However, the memory updater uses GRU which may have issues such as vanishing gradient limiting the ability to capture long term information. Also, these models have been studied for small-graphs spread over limited time duration (e.g., one month).
Considering large scale temporal graphs with evolving structures, one such application is that of sequential recommendation (SR) with decades of temporal information (1996-2018) (Zhang et al.,

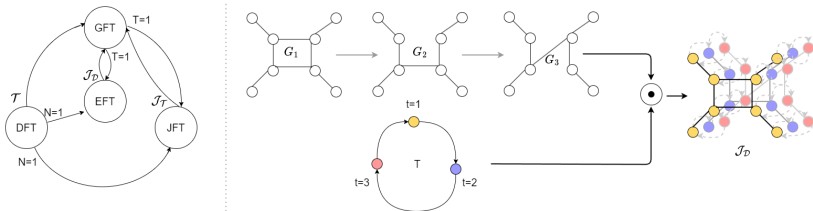

Figure 1: Left circular figure shows equivalence between *EFT* and existing transformations (DFT (Sundararajan, 2023), JFT (Loukas & Foucard, 2016), GFT (Ortega et al., 2018)). Each directed arrow (e.g, A to B), interprets as a transform simulation (transform A can be simulated by B using edge annotations). Right part shows timestamp-wise product between signals and graph structure. Here, nodes of next timestep are connected by dotted arrows to obtain the graph $\mathcal{J}_\mathcal{D}$ which can be used by GFT to simulate *EFT* (if graph is static).

2022; Huang et al., 2023). Researchers (Li et al., 2020b; Zhang et al., 2022; Jing et al., 2022) have attempted to model the sequential recommendation task as a link prediction over dynamic graphs. DGSR (Zhang et al., 2022) is a work that considers generic dynamic graphs over user-item interactions. However, the GNN-based methods described in this section including DGSR majorly employ low pass GNNs that limit the ability to model complex relations and are fundamentally restricted to focus on local neighborhood interactions (Balcilar et al., 2020).

## 3 PRELIMINARIES

**Discrete Fourier Transform** (DFT) (Sundararajan, 2023) is employed to obtain the frequency representation of a sequence of signal values sampled at equal intervals of time. Consider a signal $x$ sampled at $N$ intervals of time $t \in [0, N-1]$ to obtain the sequence $\{x_t\}$. The DFT of $x_t$ is then given by $X_k = \sum_{t=0}^{N-1} x_t e^{-i\omega_t k}$ with $\omega_t = \frac{2\pi t}{N}$. The transformed sequence $X_k$ gives the values of the signal in the frequency domain. If we represent $X$ as the vector form of the signal, we can define the DFT matrix $\mathbf{\Psi}_T$ such that $X_k = \mathbf{\Psi}_T X$.

**Graph Fourier Transform** (*GFT*) (Ortega et al., 2018) is a generalization of the Discrete Fourier Transform (DFT) to graphs. We represent a graph as $(\mathcal{V}, \mathcal{E})$ where $\mathcal{V}$ is the set of $N$ nodes and $\mathcal{E}$ represents the edges between them. Denote the adjacency matrix by $A$. $D$ is the degree matrix, defined as $(D)_i^i = \sum_j (A)_{ij}$, which is diagonal. The graph Laplacian graph is given by $\hat{L} = D - A$ and the normalized Laplacian $L$ is defined as $L = I - D^{-\frac{1}{2}} A D^{-\frac{1}{2}}$. The Laplacian($L$) has the eigendecomposition as: $L = \mathbf{\Psi}_G^* \Lambda \mathbf{\Psi}_G$. Let $X \in R^{N \times d}$ be the signal on the nodes of the graph. The Graph Fourier Transform $\hat{X}$ of $X$ is then given as: $\hat{X} = \mathbf{\Psi}_G X$.

**Pseudospectrum**: The spectrum of a graph (of N nodes) is a finite set consisting of N points $\lambda$ that form the eigenvalues of the graph's matrix representation $M$ i.e. $\{\lambda \in \mathbb{C} \mid \|(M - \lambda I)^{-1}\| = \infty\}$. Similarly we can think of the ($\epsilon$-)pseudospectrum of a graph to be the larger set (containing these N points) such that $A - \lambda I$ has the least singular value at most $\epsilon$. Formally the pseudospectrum can be defined by the set $\{\lambda \in \mathbb{C} \mid \|(M - \lambda I)^{-1}\| \geq \frac{1}{\epsilon}\}$.

**Common Notations**: We denote by $\oplus, \otimes$ the Kronecker sum, product respectively. $(M)_i^j$ refers to the $i$-th row and $j$-th column of matrix $M$. $\{.\}$ refers to a sequence, of elements, in time. $\boxtimes, \boxplus$ refer to the Kronecker product and sum respectively, applied timestep wise.

## 4 THEORETICAL FRAMEWORK: AN OPTIMIZATION PERSPECTIVE

We begin by striving for a physical interpretation of frequency for dynamic graph systems. For this, we draw inspiration from energy diffusion processes and establish similarities with the variation of signals on static graphs. Consider graph $G_t$ at time $t$ with node $n_i \in V_t$ and $n_j \overset{G_t}{\sim} n_i$ denoting the neighbors of $n_i$ at time $t$. We define a directed graph $\mathcal{J}_\mathcal{D}$ with the graphs at all timesteps taken as is and a directed edge added from a node at time $t-1$ (modulo $T$) to its

corresponding node at time $t$. For continuous time dynamic graph the previous time would be represented by $t - dt$ (modulo $T$). Let $X_{n_i,t}$ represent the energy of the signal on node $n_i$ at time $t$. The flow of energy to the node $n_i$ at time $t$ can be represented by the divergence of the gradient $(\Delta_{n_i,t}X)$ of the energy. We define the variation of the signals at time $t$ and node $n_i$ as follows:

$\|\Delta_{n_i,t}X\|_2 = \left[\sum_{n_j \overset{\mathcal{J}_\mathcal{D}}{\sim} n_i} \left(\frac{\partial X}{\partial e_{n_i n_j}}\right)^2\right]^{\frac{1}{2}} = \left[\sum_{n_j \overset{G_t}{\sim} n_i} \left(X_{n_j,t} - X_{n_i,t}\right)^2 + \left(\frac{\partial X_{n_i,t}}{\partial t}dt\right)^2\right]^{\frac{1}{2}}$, where $\frac{\partial X}{\partial e_{n_i n_j}}$ is the discrete edge derivative on the collective dynamic graph $\mathcal{J}_\mathcal{D}$. Considering $\Delta$ to be the finite difference between neighboring nodes in the joint graph, the global notion of variation $(S_p(X))$ can be given by the $p$-Dirichlet form as follows

$$S_p(X) = \frac{1}{p}\sum_{n=1}^{N}\int_{t=0}^{T}\|\Delta_{n_i,t}X\|_2^p \, dt = \frac{1}{p}\int_{t=0}^{T}\sum_{n=1}^{N}\left[\sum_{n_j \overset{G_t}{\sim} n_i}\left(X_{n_j,t} - X_{n_i,t}\right)^2 + (\delta X_{n_i,t})^2\right]^{\frac{p}{2}} dt$$

Define $L_T$ to be the Laplacian of the continuous ring graph representing the nodes at each timestep $t \in [0, T]$ and connecting consequent nodes. Let $L_{G_t}$ be the Laplacian of the sampled graph at time $t$. In the discrete case the Laplacian $L_{\mathcal{J}_\mathcal{D}}$ of $\mathcal{J}_\mathcal{D}$ can be shown to be

$$(L_{\mathcal{J}_\mathcal{D}})_i^j = (L_T \otimes I_N)_i^j + (I_T \otimes \{L_{G_t}\})_i^{j\lfloor\frac{j}{N}\rfloor} = (L_T \oplus L_{G_t})_i^{j\lfloor\frac{j}{N}\rfloor} \tag{1}$$

For the case of continuous time, this can be generalized to

$$(L_{\mathcal{J}_\mathcal{D}}) = L_T \otimes I_N + [I_T \boxtimes \{L_{G_t}\}] = [L_T \boxplus L_{G_t}] \tag{2}$$

where $\boxtimes, \boxplus$ refers to the timestep wise Kronecker product and sum respectively and $[.]$ refers to the matricization operation. In the discrete case this operation would convert $R^{NT \times T \times N} \to R^{NT \times NT}$, ordering from the last dimension first. We can now characterize the variation of signals on $J_D$ similar to static graphs by the following result:

**Lemma 1.** *(Variational Characterization of $\mathcal{J}_\mathcal{D}$) The 2-Dirichlet $S_2(X)$ of the signals $X$ on $\mathcal{J}_\mathcal{D}$ is the quadratic form of the Laplacian $L_{\mathcal{J}_\mathcal{D}}$ of $\mathcal{J}_\mathcal{D}$ i.e.*

$$S_2(X) = \int_{i=0}^{NT} vec(X)(i) \int_{j=0}^{NT} L_{\mathcal{J}_\mathcal{D}}(i,j)vec(X)(j)didj = vec(X)^T L_{\mathcal{J}_\mathcal{D}} vec(X) \tag{3}$$

This implies that $L_{\mathcal{J}_\mathcal{D}} \succeq 0$ since $S_2(X) \geq 0$, which assures us of the existence of the eigenvalue decomposition. Additionally, the value of $S_2(X)$ is lower when the signal changes slower along the dynamic graph and higher when the signal changes faster. Hence, we can define a notion of signal variation on the dynamic graph that is similar to the variation of signals on static graphs. Consequently, the eigendecomposition of $L_{\mathcal{J}_\mathcal{D}}$ characterizes signals on the dynamic graph by projecting them onto the optimizers of $S_2(X)$. This means that high-frequency components of the evolving dynamic graph represent sharply varying signals, whereas smoother signals will have a higher magnitude in the low-frequency components. From an optimization perspective, we can view the maximum frequency as the optimal value for the below equation, i.e.,

$$f_{\max} = \max_{x, \|x\|\leq 1}\int_{i=0}^{NT} x(i)\int_{j=0}^{NT} L_{\mathcal{J}_\mathcal{D}}(i,j)x(j)didj = \max_{x, \|x\|\leq 1} x^T L_{\mathcal{J}_\mathcal{D}}(i,j)x \tag{4}$$

The optimal solution $x$ provides the basis for transforming a dynamic graph signal to obtain its maximum frequency component, denoted by $f_{\max}$. We can obtain the next frequency values by optimizing equation 4 in orthogonal directions. However, this approach has an issue - the eigenvalue decomposition would have to be performed over a large number of nodes. In a real world setting of temporal graphs with $T$ timesteps, this method would have a complexity of $\mathcal{O}((NT)^3)$, which would be prohibitive considering large number of timesteps. To address this issue, we relax the objective in equation 4 to include solutions in the pseudospectrum. The solution is presented in the following result, upon which we can formulate a transformation method for temporal graphs.

**Lemma 2.** *Consider the variational form $x^T L_{\mathcal{J}_\mathcal{D}} x = \int_{i=0}^{NT} x(i)\int_{j=0}^{NT} L_{\mathcal{J}_\mathcal{D}}(i,j)x(j)didj$. The optimization problem $f = \min_{x, \|x\|\leq 1}[|x^T L_{\mathcal{J}_\mathcal{D}} x - \lambda_s| - \epsilon]_+$ has the optimal solution as $y_\omega \otimes z_l^\omega$, where $\lambda_s$ is the optimal value of equation 4, $y_\omega$ is the $\omega$-th optimal solution of the variational form of the ring graph, $z_l^t$ is the $l$-th optimal solution to the variational form of the graph at time $t$, $[s]_+ = max(s, 0)$ and $\epsilon = \mathcal{O}(\delta)$.*

# 5 CONSTRUCTING AN EVOLVING GRAPH FOURIER TRANSFORM

In the previous section, we have outlined the theoretical framework for the evolving graph Fourier transform. We also obtained a sketch of the transform as a solution to the optimization problem of the variational characterization with pseudospectrum relaxations. This enables us to obtain a simple and efficient form to compute. In this section, building upon the theoretical framework, we propose our formulation of the *Evolving Graph Fourier Transform* (*EFT*). From lemma 2, we obtain the orthogonal basis vectors of the desired transform matrix in terms of the kronecker product of the basis vectors of the Fourier Transform ($\mathbf{\Psi}_T$) and Graph Fourier Transform ($\mathbf{\Psi}_G$). Thus, lemma 2 helps us to define the *EFT* in terms of the graph and time Fourier transforms:

$$EFT(f_g, \omega) = \sum_n \mathbf{\Psi}_G(f_g, n) \int_{t=0}^T f_s(n, t) e^{-j\omega t} dt \qquad (5)$$

where $f_g, \omega$ are the graph and temporal frequency components respectively, $f_s(n, t)$ is the signal at node $n$ and time $t$. In terms of the matrix representation, the *EFT* could be expressed, using the Einstein notation (Albert et al., 1916), as a Kronecker product of DFT and GFT as $(\mathbf{\Psi}_D)_i^j = (\mathbf{\Psi}_T \otimes \{\mathbf{\Psi}_{G_t}\})_i^{j \lfloor \frac{j}{N} \rfloor}$, which when applied to the columnwise vectorized signal $f_s$ gives the transform in the spectral space.

*EFT* is one of the solutions in the pseudospectrum of $L_{\mathcal{J}_{\mathcal{D}}}$ as shown in lemma 2. There also exists other solutions and specifically considering the case where $\epsilon = 0$ we obtain the solution to the exact EVD of $L_{\mathcal{J}_{\mathcal{D}}}$. Let $\mathbf{\Psi}_{AD}$ be the matrix whose rows form the right eigenvectors of $L_{\mathcal{J}_{\mathcal{D}}}$. Since $\mathbf{\Psi}_{AD}$ is the absolute decomposition of $L_{\mathcal{J}_{\mathcal{D}}}$ we term this as *AD* for brevity. We now define error bounds between $\mathbf{\Psi}_D$ and $\mathbf{\Psi}_{AD}$.

**Theorem 1.** *Considering bounded changes in a graph $G$ with $N$ nodes over time $T$, the norm of the difference between* EFT *($\mathbf{\Psi}_D$) and AD ($\mathbf{\Psi}_{AD}$) is bounded as follows:* $\|\mathbf{\Psi}_D - \mathbf{\Psi}_{AD}\| \leq \mathcal{O}\left(N^{\frac{3}{2}} T \varepsilon(\omega_{max}, (\Delta\lambda_G)_{min}, (\Delta\lambda_J)_{min})\right) \left(\left\|\dot{L}_G\right\|\right)_{max}$ *where $(\Delta\lambda_J)_{min}$ and $(\Delta\lambda_J)_{min}$ refer to the minimum difference between the eigenvalues of matrices $L_G$ and $L_{\mathcal{J}_{\mathcal{D}}}$ respectively, $\dot{L}_G$ is the rate of change of $L_G$ and $\omega_{max} = 2\pi$ and $\varepsilon(\omega_{max}, \Delta\lambda_G, \Delta\lambda_J) = \frac{\omega_{max}^{\frac{1}{2}}}{\Delta\lambda_G} + \frac{\omega_{max}^2}{\Delta\lambda_J}$.*

The above theorem states that as the structure on the graph evolves infinitesimally, the difference between $\mathbf{\Psi}_D$ and $\mathbf{\Psi}_{AD}$ is bounded from above by the change in the graph matrix representation (laplacian/adjacency). This property is desirable since it allows us to approximate $\mathbf{\Psi}_{AD}$, which is formed by the eigendecomposition of $L_{\mathcal{J}_{\mathcal{D}}}$ and has a physical interpretation, using the defined $\mathbf{\Psi}_D$ that is easy to compute. The above bound is finite if 1) The rate of change of the graph with time is bounded. 2) The eigenvalues have a multiplicity of 1. In such cases, *EFT* characterizes signals on the dynamic graph by their proximity (projection) to the optimizers of $S_2(X)$ defined in lemma 1. The physical implication of this is that applying *EFT*, the high-frequency components correspond to sharply varying signals on a dynamic graph, while low-frequency components correspond to smoother signals. Hence, the norm of the difference between *EFT* and *AD* are bounded from above by the rate of evolution of the graphs.

For *computational purpose in real-world applications*, the sampled form of *EFT* can be obtained by sampling $T$ snapshots of the dynamic graph signal at uniform time intervals. We now get a dynamic graph $\{(\mathcal{V}_t, \mathcal{E}_t)\}, t \in \{0, T\}$ the edges ($\mathcal{E}_t$) of which by definition evolves with time. We consider the node set $\mathcal{V}$ to be fixed, i.e., no new nodes are added. All the nodes ($|\mathcal{V}| = N$) are known from the start, and the graph may contain isolated nodes. In case of node editions, we could create dummy isolated nodes with varying node signals and edge connectivity information. Without loss of generality, consider a 1-dimensional temporal signal, uniformly sampled at $T$ intervals, residing on the graph nodes. Let $X \in R^{N \times T}$ represent the temporal signal on the graph nodes. The Fourier transform (DFT) (with DFT matrix $\mathbf{\Psi}_T$) independently for each node is $DFT(X) = X\mathbf{\Psi}_T^\top$. Further, the *GFT* for the graph $G_t \equiv (\mathcal{V}_t, \mathcal{E}_t)$ at time $t$ is given as $GFT(X_t) = \mathbf{\Psi}_{G_t} X_t$, where $X_t \in R^N$ is the signal on the graph nodes at time $t$. In order to compute the dynamic graph transform along the graph domain as well as the temporal dimension, we can *collectively* perform both the operations.

Consider $\{\mathbf{\Psi}_{G_t}\} \in R^{N \times N \times T}$ as the tensor containing the graph Fourier basis at each timestep. Then using Einstein notation (Albert et al., 1916), we write *EFT* as

$$\left(\mathbf{EFT}(\{G_t\}; X)\right)^j_i = \left(\mathbf{\Psi}_{G_t} X\right)^{kk}_i \left(\mathbf{\Psi}^\top_T\right)^j_k \tag{6}$$

where $i, j, k$ are tensor indices. Next, we aim to define a transformation matrix for *EFT* as in DFT and GFT. For this we make use of the Kronecker product ($\otimes$) between tensors. We then get the matrix form of *EFT* as the following expression:

$$\left(\mathbf{EFT}(\{G_t\}; X)\right)^j_i = \left(\hat{X}_G\right)^j_i = \left(\mathbf{\Psi}_{G_t} X\right)^{kk}_i \left(\mathbf{\Psi}^\top_T\right)^j_k = \left(\mathbf{\Psi}_T \otimes \{\mathbf{\Psi}_{G_t}\}\right)^{km}_{(j*N+i)} x_k \tag{7}$$

Thus, we have $\hat{x}_{j*N+i} = \left(\mathbf{\Psi}_T \otimes \{\mathbf{\Psi}_{G_t}\}\right)^{km}_{(j*N+i)} x_k$ or $\hat{x} = \mathbf{\Psi}_D x$. In the above equations, $\hat{X}_G$ is the *EFT* of signal $X$ over dynamic graph $\{G_t\}$, $x, \hat{x} \in R^{NT}$ are the columnwise vectorized form of $X, \hat{X}_G \in R^{N \times T}$ and $m = \lfloor \frac{k}{N} \rfloor$. $\mathbf{\Psi}_D \in R^{NT \times NT}$ is the *EFT* matrix over dynamic graph $\{G_t\}$ with $(\mathbf{\Psi}_D)^j_i = (\mathbf{\Psi}_T \otimes \mathbf{\Psi}_G)^{j \lfloor \frac{i}{N} \rfloor}_i$.

We remark from equation 6 of *EFT* , that the following desirable properties (over the exact eigen-decompostion of the joint laplacian) are satisfied: 1) The equation imparts interpretibility to the frequency components, whether belonging to the time or vertex domain, as compared to the exact eigendecomposition. This is possible because we are able to decompose the transform into the individual transforms of each domain. 2) The transform equation is computationally efficient as compared to the exact eigendecomposition of the joint laplacian. Specifically *EFT* reduces the computational complexity for the dynamic graph ($T$ timesteps) from a factor of $\mathcal{O}(T^3)$ to $\mathcal{O}(T + T\log(T))$.

Having derived the *EFT* transform, we state and prove its properties in the appendix C. The illustration between *EFT* and other transforms is in Figure 1. The figure shows transforms (*GFT, JFT, DFT, EFT)* in a circle, and arrows from one transform to the next indicate that the source transform can be obtained by the destination transform using the simulation annotated on the edges. For example, the *GFT* of a ring graph ($\mathcal{T}$) gives the *DFT* and thus the DFT can be simulated by *GFT* using graph $\mathcal{T}$. Similarly *DFT* can be simulated by *EFT* when the number of nodes $N = 1$. Also the *GFT* of the temporal ring of a static graph (topologically equivalent to a torus), where the nodes and edges remain constant with time, gives the *EFT* and vice versa (when time $T = 1$). However when the graph structure changes with time *GFT* cannot be used to simultae *EFT* . Thus, we can also look at the *EFT* as a generalization of the previous transforms. We briefly explain the task specific implementation of these modules in the below subsection and focus more on the representations and results in the following sections.

## 5.1 Implementation Details

Having obtained the representations using the proposed transform, we intend to perform filtering in spectral space for dynamic graphs. Since our idea is to perform collective filtering along the vertex and temporal domain in *EFT*, we need two modules to compute $\mathbf{\Psi}_{G_t}$ (vertex aspect) and $\mathbf{\Psi}_T$ (temporal aspect), respectively, in equation 6 of *EFT*. We now briefly explain these modules with details in appendix D.2.

**Filtering along the Vertex Domain:** This module computes the convolution matrix $\mathbf{\Psi}_{G_t}$ in equation 6. The frequency response of the desired filter is approximated as $\hat{\Lambda}_l = \sum_{k=0}^{O_f} c_k T_k(\tilde{\Lambda})$, where $O_f$ is the polynomial/filter order, $T_k$ is the Chebyshev polynomial basis, $\tilde{\Lambda} = \frac{2\Lambda}{\lambda_{max}} - I$, $\lambda_{max}$ is the maximum eigenvalue and $c_k$ is the corresponding *filter coefficients*. The convolution of the graph signal $X$ with the filter ($X * \Lambda_l$) gives the desired filter response in the vertex domain.

**Filtering along the Temporal Domain:** After performing filtering in the vertex domain, we aim to filter over the temporal signals using $\mathbf{\Psi}_T$ as in equation 6. Formally, let $X_t \in R^d$ be the signal of a node at time $t$. Let $X = \{X_t\} \in R^{T \times d}$ be the time ordered matrix of embeddings of the node. This is converted to the frequency domain ($\hat{X} \in R^{T \times d}$) using the matrix $\mathbf{\Psi}_T$ as $\hat{X} = \mathbf{\Psi}_T X$. Then we multiply $\hat{X}$ element-wise by a temporal filter $F_T \in R^{T \times d}$ to obtain the filtered signal $\hat{X}_f = F_T \odot \hat{X}$ which is then converted back to the temporal domain by using the inverse transform $\mathbf{\Psi}^*_T$ to get $X_f = \mathbf{\Psi}^*_T \hat{X}_f$. $X_f$ is the filtered signal in the time-vertex domain of the dynamic graph.

## 6 EXPERIMENTAL SETUP

**Model Implementation and Datasets:** Considering *EFT* is a spectral transform, we need a base model to induce *EFT* in it. We select transformer as the base model inspired from (Zhou et al., 2022; Bastos et al., 2022) that induce learnable filters into a vanilla transformer for downstream tasks (implementation is inspired from (Zhou et al., 2022), hence, details are in appendix). To illustrate the efficacy of the representations obtained from *EFT*, we consider eight datasets. We name our model *EFT-T*. The first three (Amazon Beauty, Games, CD in Table 3) are large continuous time dynamic graph datasets from sequential recommendation (SR) setting (Huang et al., 2023), spread over two decades. We inherit these datasets, dynamic graph construction process in SR setting, and metric from (Zhang et al., 2022). Other datasets (Pareja et al., 2020) (UCI, AS, SBM, Elliptic, Brain) are standard (discrete) dynamic graph datasets to understand the generalizability of our method and contain a sequence of time-ordered graphs. Details on datasets, metrics, and experiment settings are in Appendix (cf., Table 4). Experiment code and associated datasets are on Github: `https://github.com/ansonb/EFT`.

**Baselines:** We use baselines depending on the experiment setting for fairness. For SR link prediction, we use strong baselines from previous best (Zhang et al., 2022): BPR-MF (Rendle et al., 2009), FPMC (Rendle et al., 2010), GRU4Rec+ (Hidasi & Karatzoglou, 2018), Caser (Tang & Wang, 2018), SASRec (Kang & McAuley, 2018), HGN (Ma et al., 2019), TiSASRec (Li et al., 2020a), SRGNN (Wu et al., 2019), HyperRec (Wang et al., 2020), FMLPRec (Zhou et al., 2022), and DGSR (Zhang et al., 2022). For link prediction, node classification on discrete dynamic graph datasets, we rely on state of the art approaches of this setting (Xiang et al., 2022): GCN (Kipf & Welling, 2017), GAT (Veličković et al., 2018), GCN-GRU (Pareja et al., 2020), DynGEM (Goyal et al., 2017), GAEN (Shi et al., 2021), EvolveGCN (Pareja et al., 2020), dyngraph2vec (dg2vec) (Goyal et al., 2020).

## 7 RESULTS AND DISCUSSION

This section reports the various experiment results, supporting our theoretical contributions.

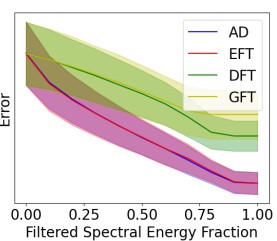

Figure 2: Reconstruction error on noisy synthetic data.

**Denoising and reconstruction on synthetic dataset with perturbation:** Here, we aim to study whether *EFT* can better filter out noise from a dynamic graph than DFT (Sundararajan, 2023) and GFT (Ortega et al., 2018). The graphs are generated by sampling edge weights from a random normal distribution and evolved by perturbing the edge weights from the previous timestep. The graph signals are sampled from the eigenvectors of the graphs at each timestep, while the temporal signals are sampled from a sinusoidal signal. To add an element of complexity and realism, noise is induced along both the graph vertex and time signals (details in appendix D). As a result, the dynamic graph signals evolve with time while being induced with noise along both dimensions. We hypothesize that using *EFT*, which transforms collectively across time and vertex dimensions, will result in better denoising and signal reconstruction compared to using GFT or DFT, which only performs filtering in one dimension. Our hypothesis is confirmed in Figure 2, which shows a decrease in error as the spectral energy of the signal is preserved while noise is filtered. Moreover, *EFT* yields comparable results to absolute transform (*AD*) while requiring less computational resources.

**Compactness of *EFT*:** Compaction refers to the ability of the transform to summarize the data compactly. A transform with good compaction is desirable as it summarizes the signals well in the frequency components, which can be used for efficient processing by downstream models. In this experiment, we verify the compaction properties of the proposed transform for the time-vertex frequencies on the temporal mesh graphs (Grassi et al., 2017) concerning GFT and DFT. In order to test this, we remove varying percentile of the frequency components from the transformed frequency domain of signal $X$. We then apply the inverse transform to obtain the signal $X_r$. We plot the error $\frac{\|X-X_r\|_F}{\|X\|_F}$ vs the percentile of components removed. From figure 3a, 3b we can see that *EFT* has a lower error and better compaction and thus is able to summarize the data better than the baselines that only transform along a single dimension of vertex or time.

Table 1: For link prediction on large temporal graphs of sequential recommendation setting, table shows our model comparison (EFT-T) on the metrics *Recall@10* and *NDCG@10*. The best results are shown in boldface. The second best result has been underlined. The improvement of our method over the best-performing baseline is statistically significant with p < 0.05.

| | GRU4Rec+ | Caser | SASRec | HGN | TiSASRec | FMLPRec | SRGNN | HyperRec | DGSR | EFT-T |
|---|---|---|---|---|---|---|---|---|---|---|
| | | | | | Recall@10 | | | | | |
| *Beauty* | 43.98 | 42.64 | 48.54 | 48.63 | 46.87 | 47.47 | 48.62 | 34.71 | 52.40 | **53.23** |
| *Games* | 67.15 | 68.83 | 73.98 | 71.42 | 71.85 | 73.62 | 73.49 | 71.24 | 75.57 | **77.78** |
| *CDs* | 67.84 | 61.65 | 71.32 | 71.42 | 71.00 | 72.41 | 69.63 | 71.02 | 72.43 | **75.42** |
| | | | | | NDCG@10 | | | | | |
| *Beauty* | 26.42 | 25.47 | 32.19 | 32.47 | 30.45 | 32.38 | 32.33 | 23.26 | 35.90 | **37.10** |
| *Games* | 45.64 | 45.93 | 53.60 | 49.34 | 50.19 | 51.26 | 53.35 | 48.96 | 55.70 | **58.65** |
| *CDs* | 44.52 | 45.85 | 49.23 | 49.34 | 48.97 | 53.31 | 48.95 | 47.16 | 51.22 | **54.99** |

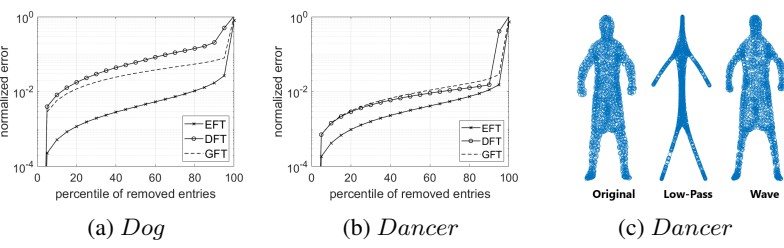

(a) *Dog*     (b) *Dancer*     (c) *Dancer*

Figure 3: Representations on dynamic mesh datasets. Left (a,b): Reconstruction error on the datasets illustrating the compactness of *EFT* . Right (c): Illustration of filtering using *EFT* on the dynamic mesh of a Dancer.

**Illustration of filtering on temporal mesh** Figure 3c shows an example of collective filtering of a dynamic mesh representing a dancer (Grassi et al., 2017). Similar to Grassi et al. (2017), we implement the following filters: (a) a low-pass filter that jointly attenuates high frequency components of the dynamic graph, and (b) a wave filter whose frequency response is described in Eq. (19) of Grassi et al. (2017). The former filter gives us the frame of the mesh with stiff manoeuvers, whereas the fluid filter produces fluid movements. This experiment shows that *EFT* can enhance the frequency components non-linearly. This also hints towards why *EFT* performs better on evolving temporal graphs in subsequent experiments.

**Performance comparison on (continuous) large-scale temporal graph datasets:** The results on the large-scale SR datasets are in Table 1 and *EFT-T* outperforms baselines on all datasets. We note that our gains to the best baseline are higher in CDs, followed by the Games and Beauty dataset. We observe that as the density of the graph and length of sequences in the data increases (e.g., CD dataset), the performance of *EFT-T* enhances. We believe that as graph density increases, higher-order connections may encompass

Table 2: Results for Link Prediction (UCI, SBM, AS) and Node Classification (Brn, Ell) tasks. Best values are in bold and second bests are underlined.

| Datasets | SBM | | UCI | | AS | | Ell | Brn |
|---|---|---|---|---|---|---|---|---|
| Metrics | MAP | MRR | MAP | MRR | MAP | MRR | F1 | F1 |
| GCN | 0.189 | 0.014 | 0.000 | 0.047 | 0.002 | 0.181 | 0.434 | 0.232 |
| GAT | 0.175 | 0.013 | 0.000 | 0.047 | 0.020 | 0.139 | 0.451 | 0.121 |
| DynGEM | 0.168 | 0.014 | 0.021 | 0.106 | 0.053 | 0.103 | 0.502 | 0.225 |
| GCN-GRU | 0.190 | 0.012 | 0.011 | 0.098 | 0.071 | 0.339 | 0.575 | 0.186 |
| dg2vec v1 | 0.098 | 0.008 | 0.004 | 0.054 | 0.033 | 0.070 | 0.464 | 0.191 |
| dg2vec v2 | 0.159 | 0.012 | 0.020 | 0.071 | 0.071 | 0.049 | 0.442 | 0.215 |
| GAEN | 0.1828 | 0.008 | 0.000 | 0.049 | 0.130 | 0.051 | 0.492 | 0.205 |
| EGCN-H | 0.195 | 0.014 | 0.013 | 0.090 | 0.153 | 0.363 | 0.391 | 0.225 |
| EGCN-O | 0.200 | 0.014 | 0.027 | 0.138 | 0.114 | 0.275 | 0.544 | 0.192 |
| LED-GCN | 0.196 | 0.015 | 0.032 | 0.163 | 0.193 | 0.469 | 0.471 | 0.261 |
| LED-GAT | 0.182 | 0.012 | 0.026 | 0.149 | 0.233 | 0.384 | 0.503 | 0.150 |
| EFT-T | **0.250** | **0.024** | **0.055** | **0.181** | **0.672** | **0.689** | **0.616** | **0.308** |

noisy relations, a challenge conventional baselines struggle to filter out, whereas our method effectively handles this noise. Also, *EFT* effectively captures global interactions as it considers the temporal aspect in the collective filtering module. Furthermore, compared to the FMLPRec model that induces DFT into a transformer, *EFT-T* performs significantly better, concluding the necessity of capturing evolving spectra of temporal graphs. We also note that among the graph-based methods, SRGNN only considers connectivity information from the sequence graph, whereas HyperRec uses higher-order connectivity information. This indicates that not using the graph information effectively hampers performance but using higher-order connectivity without filtering to remove noise also degrades the results.

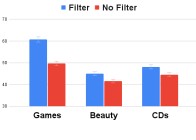 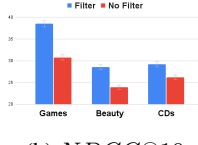 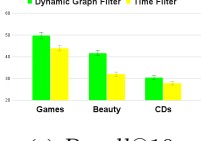 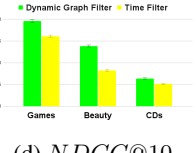

(a) $Recall@10$    (b) $NDCG@10$    (c) $Recall@10$    (d) $NDCG@10$

Figure 4: Effect of inducing 1) semantic noise in embeddings with and without filters (a-b) 2) structural noise in the form of graph perturbations with and without graph filters (c-d), on the performance of *EFT* . We consider large-scale SR setting.

**Performance comparison on discrete temporal graph datasets:** Table 2 summarizes link prediction and node classification results. Across datasets, our model significantly outperforms all baselines, which focus on learning local dependencies. It illustrates our framework's effectiveness in filtering noise and amplifying useful signals in evolving temporal graphs.

**Effectiveness of filtering module (Figure 4):** Our approach focuses on capturing useful frequencies along vertex and time dimensions collectively while filtering the noise. Hence, in this experiment, we aim to understand the effectiveness of the filters along both graphs (vertex) and time dimension in the presence of *explicitly added noise*.

Firstly, we induce *semantic noise* into the system by adding a random vector (sampled from a normal distribution) to the node embeddings. Then, we run experiments on our model with and without learnable collective graph-time filters. To ensure a fair comparison, we keep the parameters in both models the same and simulate the no-filter configuration by using a uniform distribution for the frequency response (all-pass filter). In the presence of noise, the performance of configuration with filters is much better ($p < 0.01$) than that without any filtering. Next, we induce *structural noise* into the system by adding random nodes/edges. We observe that on inducing structural noise, the performance of the con-

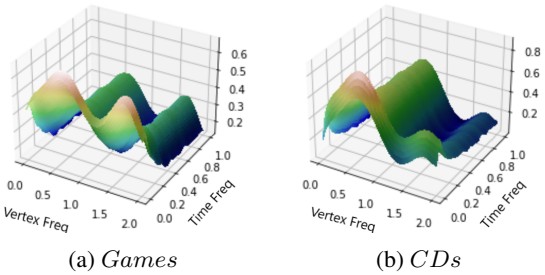

(a) $Games$    (b) $CDs$

Figure 5: Filter frequency responses learnt by *EFT* on dynamic graph datasets. The x-axis shows the vertex frequency (0-2), y axis shows the normalized temporal frequency and z axis shows the magnitudes of the normalized frequency response.

figuration with graph filters is statistically better ($p < 0.01$ using a paired t-test) compared to the one without, confirming that collective filtering is needed to be robust to structural noise in dynamic graphs. Additionally, we plotted the filter frequency responses of *EFT* on the Games and CDs datasets in Figure 5. The figure shows dominating low-frequency response and higher-frequency components, indicating global aggregation for the long-range interactions.

## 8 CONCLUSION

In this paper, we introduce a novel approach to transform temporal graphs into the frequency domain, grounded on theoretical foundations. We propose pseudospectrum relaxations to the variational objective obtaining a simplified transformation, making it computationally efficient for real-world applications. We show that the error between the proposed transform and the exact solution to the variational objective is bounded from above and study its properties. We further demonstrate the practical effectiveness for temporal graphs. In the current scope, we do not consider generic signed and directed graphs. To mitigate this, we suggest future works explore generalizing the Laplacian and the resulting transform to such graphs, leveraging techniques proposed in (Mercado et al., 2016; Cucuringu et al., 2021). Our work opens up new possibilities for dynamic graph analysis and representation learning, and we encourage researchers to explore potential of *EFT* as a spectral representation of the evolving graph in downstream graph representation learning models.

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

# A    PRELIMINARIES

Here we give an extended discussion of the preliminaries which could not be accommodated in the main paper due to space constraints.

## A.1    DISCRETE FOURIER TRANSFORM

The Discrete Fourier Transform (DFT) is used to obtain the frequency representation of a sequence of signal values sampled at equal intervals of time. The magnitude and phase of the frequency components are obtained by multiplying the signal values by complex sinusoids of the respective frequencies. Consider a signal $x$ sampled at $N$ intervals of time $t \in [0, N-1]$ to obtain the sequence $\{x_t\}$. The DFT of $x_t$ is then given by,

$$X_k = \sum_{t=0}^{N-1} x_t e^{-i\omega_t k}, \quad \omega_t = \frac{2\pi t}{N} \tag{8}$$

The transformed sequence $X_k$ gives the values of the signal in the frequency domain. If we represent $X$ as the vector form of the signal we can define the DFT matrix $\mathbf{\Psi}_T$ such that $X_k = \mathbf{\Psi}_T X$. Thus $X_k$ is the complex valued spectrum of $\{x_t\}$ at frequency $\omega_t$. We can perform filtering by removal of noisy frequencies in this spectral domain. As required by signal processing applications we can then obtain the signal sequence in the time domain from the frequency domain $\{X_k\}$ using the Inverse Discrete Fourier Transform (IDFT) as,

$$x_t = \frac{1}{N} \sum_{k=0}^{N-1} X_k e^{i\omega_k t}, \quad \omega_k = \frac{2\pi k}{N} \tag{9}$$

## A.2    GRAPH FOURIER TRANSFORM

Graph Fourier Transform (GFT) is a generalization of the Discrete Fourier Transform to graphs. We represent a graph as $(\mathcal{V}, \mathcal{E})$ where $\mathcal{V}$ is the set of $N$ nodes and $\mathcal{E}$ represents the edges between them. Denote the adjacency matrix by $A$. In the setting of an undirected graph $A$ would be a symmetric matrix. $D$ is the degree matrix, defined as $(D)_{ii} = \sum_j (A)_{ij}$, which is diagonal. The Laplacian of the graph is given by $\hat{L} = D - A$ and the normalized Laplacian $L$ is defined as $L = I - D^{-\frac{1}{2}} A D^{-\frac{1}{2}}$. The Laplacian($L$) can be decomposed into its orthogonal basis, namely the eigenvectors and eigenvalues as:$L = \mathbf{\Psi}_G^* \Lambda \mathbf{\Psi}_G$, where U is an $N \times N$ matrix whose columns are the eigenvectors corresponding to the eigenvalues $\lambda_1, \lambda_2, \ldots, \lambda_N$ and $\Lambda = \mathrm{diag}([\lambda_1, \lambda_2, \ldots, \lambda_n])$. Let $X \in R^{N \times d}$ be the signal on the nodes of the graph. The Fourier Transform $\hat{X}$ of $X$ is then given as: $\hat{X} = \mathbf{\Psi}_G X$. Similarly, the inverse Fourier Transform is defined as: $X = \mathbf{\Psi}_G^* \hat{X}$. Note $\mathbf{\Psi}_G^*$ is the transposed conjugate of $\mathbf{\Psi}_G$. By the convolution theorem (Blackledge, 2005), the convolution of the signal $X$ with a filter G having its frequency response as $\hat{G}$ is given by (below, $m$ represents the $m^{th}$ node in the graph, $\mathbf{\Psi}_{G_k}$ represents the $k^{th}$ eigenvector or column of $\mathbf{\Psi}_G$):

$$(X * G)(m) = \sum_{k=1}^{N} \hat{X}(k) \hat{G}(k) \mathbf{\Psi}_{G_k}(m)$$
$$(X * G)(m) = \sum_{k=1}^{N} (\mathbf{\Psi}_G X)(k) \hat{G}(k) \mathbf{\Psi}_{G_k}^*(m) \tag{10}$$
$$X * G = \mathbf{\Psi}_G^* \hat{G} \mathbf{\Psi}_G X$$

Note that a sequence can be considered as a grid graph and for this graph the GFT specializes to the DFT i.e $\mathbf{\Psi}_T = \mathbf{\Psi}_G$.

# B    THEORETICAL PROOFS

In this section we outline the proofs stated in the main paper and briefly discuss the implications etc. that could not be accommodated in the main paper due to space constraints. For completeness we restate the results.

**Lemma 1.** (*Variational Characterization of $\mathcal{J}_\mathcal{D}$*) *The* 2-*Dirichlet* $S_2(X)$ *of the signals* $X$ *on* $\mathcal{J}_\mathcal{D}$ *is the quadratic form of the Laplacian* $L_{\mathcal{J}_\mathcal{D}}$ *of* $\mathcal{J}_\mathcal{D}$ *i.e.*

$$S_2(X) = vec(X)^T L_{\mathcal{J}_\mathcal{D}} vec(X)$$

*Proof.* The $p$-Dirichlet form is given by

$$S_p(X) = \frac{1}{p} \sum_{n=1}^{N} \int_{t=0}^{T} \|\Delta_{n_i,t} X\|_p^2$$

$$= \frac{1}{p} \sum_{t=1}^{T} \sum_{n=1}^{N} \left[ \sum_{n_j \overset{G_t}{\sim} n_i} \left( X_{n_j,t} - X_{n_i,t} \right)^2 + \left( \frac{\partial X_{n_i,t}}{\partial t} dt \right)^2 \right]^{\frac{p}{2}}$$

Thus the 2-Dirichlet form is

$$S_2(X) = \frac{1}{2} \sum_{n=1}^{N} \int_{t=0}^{T} \|\Delta_{n_i,t} X\|_2^2$$

$$= \frac{1}{2} \int_{t=0}^{T} \sum_{n=1}^{N} \left[ \sum_{n_j \overset{G_t}{\sim} n_i} \left( X_{n_j,t} - X_{n_i,t} \right)^2 + \left( \frac{\partial X_{n_i,t}}{\partial t} dt \right)^2 \right]$$

$$= \frac{1}{2} \left( \int_{t=0}^{T} \sum_{n=1}^{N} \sum_{n_j \overset{G_t}{\sim} n_i} \left( \delta X_{n_i,t} \right)^2 \right)$$

$$+ \frac{1}{2} \left( \int_{t=0}^{T} \sum_{n=1}^{N} \left( X_{n_i,t-dt} - X_{n_i,t} \right)^2 \right)$$

We consider the above sum in two parts. Taking the first part we have

$$\frac{1}{2} \int_{t=0}^{T} \sum_{n=1}^{N} \sum_{n_j \overset{G_t}{\sim} n_i} \left( X_{n_j,t} - X_{n_i,t} \right)^2$$

$$= \frac{1}{2} \int_{t=0}^{T} \sum_{n=1}^{N} \sum_{n_j \overset{G_t}{\sim} n_i} \left( X_{n_j,t}^2 - 2 * X_{n_j,t} * X_{n_i,t} + X_{n_i,t}^2 \right)$$

$$= \frac{1}{2} \int_{t=0}^{T} \sum_{n=1}^{N} \sum_{n_j \overset{G_t}{\sim} n_i} \left( X_{n_j,t}^2 - X_{n_j,t} * X_{n_i,t} - X_{n_j,t} * X_{n_i,t} + X_{n_i,t}^2 \right)$$

$$= \frac{1}{2} \int_{t=0}^{T} \sum_{n=1}^{N} \sum_{n_j \overset{G_t}{\sim} n_i} \left( X_{n_j,t} \left( X_{n_j,t} - X_{n_i,t} \right) + X_{n_i,t} \left( X_{n_i,t} - X_{n_j,t} \right) \right)$$

$$= \frac{1}{2} \int_{t=0}^{T} \sum_{n=1}^{N} 2 * \sum_{n_j \overset{G_t}{\sim} n_i} \left( X_{n_j,t} \left( X_{n_j,t} - X_{n_i,t} \right) \right)$$

$$= \int_{t=0}^{T} \sum_{n=1}^{N} \sum_{n_j \overset{G_t}{\sim} n_i} \left( X_{n_j,t} \left( X_{n_j,t} - X_{n_i,t} \right) \right)$$

$$= vec(X)^\top (I_T \otimes L_{G_t}) vec(X)$$

Considering the second part which is the ring graph along the time dimension we have

$$\frac{1}{2}\left(\int_{t=0}^{T}\sum_{n=1}^{N}\left(X_{n_i,t-dt}-X_{n_i,t}\right)^2\right)$$

$$=\frac{1}{2}\left(\int_{t=0}^{T}\sum_{n=1}^{N}X_{n_i,t-dt}^2-2*X_{n_i,t-dt}*X_{n_i,t}+X_{n_i,t}^2\right)$$

$$=\frac{1}{2}\left(\int_{t=0}^{T}\sum_{n=1}^{N}2X_{n_i,t-dt}^2-2*X_{n_i,t-dt}*X_{n_i,t}\right)\quad\ldots\text{Redistributing terms}$$

$$=\int_{t=0}^{T}\sum_{n=1}^{N}X_{n_i,t-dt}^2-X_{n_i,t-dt}*X_{n_i,t}$$

$$=\int_{t=0}^{T}\sum_{n=1}^{N}X_{n_i,t-dt}\left(X_{n_i,t-dt}-X_{n_i,t}\right)$$

$$=vec(X)^{\top}(L_T\otimes I_N)vec(X)$$

Combining the results of the 2 parts we get the below result

$$S_2(X)=vec(X)^{\top}(I_T\otimes L_{G_t})vec(X)+vec(X)^{\top}(L_T\otimes I_N)vec(X)$$

$$=vec(X)^{\top}\left(I_T\otimes L_{G_t}+L_T\otimes I_N\right)vec(X)$$

$$=vec(X)^{\top}L_{\mathcal{J}_{\mathcal{D}}}vec(X)$$

as required.

$\square$

This implies that $L_{\mathcal{J}_{\mathcal{D}}}\succeq 0$. We can see that slower the changes in the signals along the dynamic graph smaller the value of $S_2(X)$ and vice versa. Thus we have a notion of variation of signals on the dynamic graph similar to the case of static graphs. The eigen decomposition of $L_{\mathcal{J}_{\mathcal{D}}}$ therefore characterizes signals on the dynamic graph by its projection to the optimizers of $S_2(X)$. In other words, high collective dynamic graph frequency components inform of the presence of sharply varying signals and smoother signals will have higher magnitude in the low frequency components. Next we provide a solution to the relaxed pseudospectrum objective in 2.

**Lemma 2.** *Consider the variational form* $x^T L_{\mathcal{J}_{\mathcal{D}}}x=\int_{i=0}^{NT}x(i)\int_{j=0}^{NT}L_{\mathcal{J}_{\mathcal{D}}}(i,j)x(j)didj$. *The optimization problem* $f=\min\limits_{x,\|x\|\leq 1}[|x^T L_{\mathcal{J}_{\mathcal{D}}}x-\lambda_s|-\epsilon]_+$ *has the optimal solution as* $y_\omega\otimes z_l^\omega$, *where* $\lambda_s$ *is the optimal value of equation 4,* $y_\omega$ *is the* $\omega$-th *optimal solution of the ring graph,* $z_l^t$ *is the* l-th *optimal solution of the graph at time t and* $\epsilon=\mathcal{O}(\delta)$.

*Proof.* We begin by considering the variational characterization of $L_{\mathcal{J}_{\mathcal{D}}}$ which is given by the below equation

$$\lambda_s=\max\limits_{x,\|x\|\leq 1}\int_{i=0}^{NT}x(i)\int_{j=0}^{NT}L_{\mathcal{J}_{\mathcal{D}}}(i,j)x(j)didj\max\limits_{x,\|x\|\leq 1}x^T L_{\mathcal{J}_{\mathcal{D}}}x\qquad(11)$$

Note that in the objective, $x^T L_{\mathcal{J}_{\mathcal{D}}}x$ is convex since $L_{\mathcal{J}_{\mathcal{D}}}\succeq 0$ and thus $\nabla^2\left(x^T L_{\mathcal{J}_{\mathcal{D}}}x\right)=2L_{\mathcal{J}_{\mathcal{D}}}\succeq 0$. Also, we can check that $\|x\|^2\leq 1$ is convex. Thus applying the KKT conditions (Gass & Fu, 2013) to the lagrangian $L=x^T L_{\mathcal{J}_{\mathcal{D}}}x+\lambda(\|x\|^2-1)$, we get the below equation

$$L_{\mathcal{J}_{\mathcal{D}}}x=\lambda_s x\qquad(12)$$

We recognize from the above equation that $\lambda_s$ is the eigenvalue of $L_{\mathcal{J}_{\mathcal{D}}}$ and x is the corresponding eigenvector. However the computation of this exact solution is computationally costly and here we are ineterested in finding an efficient form of the solution to the objective with the pseudospectrum relaxation. As already seen, the pseudospectrum can be defined by the set $\{\lambda\in\mathbb{C}\mid\|(L_{\mathcal{J}_{\mathcal{D}}}-\lambda I)^{-1}\|\geq\frac{1}{\epsilon}\}$ or equivalently $\{\lambda\in\mathbb{C}\mid\|(L_{\mathcal{J}_{\mathcal{D}}}-\lambda I)\|\leq\epsilon\}$, where

$\|.\|$ is the operator norm. Thus we have that for the pseudospectrum, there exists a unit vector $v$ such that $|(L_{\mathcal{J}_{\mathcal{D}}} - \lambda I)v| \leq \epsilon$ and so $|\lambda_s - \lambda| \leq \epsilon$. This shows that the $\epsilon-$ neighborhood of the spectrum of $L_{\mathcal{J}_{\mathcal{D}}}$ is contained in the pseudospectrum i.e. if $\lambda$ is in the pseudospectrum of $L_{\mathcal{J}_{\mathcal{D}}}$ it is in the $\epsilon-$ neighborhood of $\lambda_s$. We would now like to find a solution residing in the pseudospectrum of $L_{\mathcal{J}_{\mathcal{D}}}$.

We have $\{L_{G_t}\} \in R^{N \times N \times T}$ to be the Laplacian of the graphs at each timestep with eigenvalues $\lambda_i^t$ where $i \in N, t \in [0, T]$. $L_T \in R^{T \times T}$ be the Laplacian of the time adjacency matrix with eigenvalues $\mu_j$ where $j \in T$. The Laplacian of the collective graph $\mathcal{J}_{\mathcal{D}}$ is expressed as

$$L_{\mathcal{J}_{\mathcal{D}}} = L_T \oplus \{L_{G_t}\} = L_T \otimes I_N + [I_T \boxtimes \{L_{G_t}\}]$$

In the above equation, $\boxtimes$ is the timestep wise Kronecker product and operator $[.]$ represents the vectorization. If $T$ is discrete this vectorization can be thought of as a reordering from $R^{NT \times T \times N} \to R^{NT \times NT}$. Consider $a_1, a_2, \ldots a_p$ to be the linearly independent right eigenvectors of $L_T$ and $b_1^t, b_2^t, \ldots b_{q_t}^t$ to be the linearly independent right eigenvectors of $L_{G_t}$. Consider the vector $y = [a_k \boxtimes \{b_l^t\}]$, where $\{b_l^t\}$ represents the set of eigenvectors of Laplacian at time $t$ i.e. $L_{G_t}$ and the operator $\boxtimes$ is again timestep wise followed by vectorization. Then we have

$$
\begin{aligned}
L_{\mathcal{J}_{\mathcal{D}}} y &= L_T \otimes I_N y + [I_T \boxtimes \{L_{G_t}\}] y \\
&= (L_T \otimes \{I_N\})[a_k \boxtimes \{b_l^t\}] + [I_T \boxtimes \{L_{G_t}\}][a_k \boxtimes \{b_l^t\}] \\
&= (L_T \otimes \{I_N\} \square [a_k \boxtimes \{b_l^t\}]) + (I_T \otimes \{L_{G_t}\} \square [a_k \boxtimes \{b_l^t\}]) \\
&= [L_T a_k \boxtimes \{I_N\} \square \{b_l^t\}] + [I_T a_k \boxtimes \{L_{G_t} \square \{b_l^t\}\}] \\
&= (\mu_k [a_k \boxtimes \{b_l^t\}]) + [a_k \boxtimes \{\lambda_i^t \{b_l^t\}\}] \\
&= (\mu_k [a_k \boxtimes \{b_l^t\}] + [a_k \boxtimes \{\lambda_i^t \{b_l^t\}\}] \\
&= ([a_k \boxtimes \{b_l^t\}] diag(\{\mu_k\}) + [a_k \boxtimes \{b_l^t\} diag(\{\lambda_i^t\})]) \\
&= ([a_k \boxtimes \{b_l^t\}] diag(\{\mu_k + \lambda_i^t\}))
\end{aligned}
$$

where $\square$ indicates timestep (column) wise product and $diag(.)$ operator converts a vector to a diagonal matrix. Now considering the graph at the 0-th timestep having eigenvalue $\lambda_l^0$, we are interested in verifying the pseudospectrum condition for $\mu_k + \lambda_l^0\}$. We thus have to find the upper bound for $\|L_{\mathcal{J}_{\mathcal{D}}} - (\mu_k + \lambda_l^0)I\|$.

In order to bound the above expression we consider the vector $y = [a_k \boxtimes \{b_l^t\}]$. We have from the above equations,

$$\|L_{\mathcal{J}_{\mathcal{D}}} y - (\mu_k + \lambda_l^0)y\| = ([a_k \boxtimes \{b_l^t\}] diag(\{\mu_k + \lambda_l^t - (\mu_k + \lambda_l^0)\})) \tag{13}$$

We would like to study the rate of change of the eigenvalues as the graph changes. Consider a normal matrix $A$ of which the eigenvectors $v_1, \ldots, v_n$ form a basis of $\mathbb{C}^n$. Also we consider $w_1, \ldots, w_n$ be the dual basis, i.e. $w_j^* v_k = \delta_{jk}$ for all $1 \leq j, k \leq n$, where $\delta_{jk}$ is the Kronecker delta and

$$\delta_{jk} = \begin{cases} 1, & \text{if } j = k \\ 0, & \text{otherwise} \end{cases}$$

Since the eigenvectors form a basis we can represent any vector $u$ as a linear combination of $v_1, \ldots, v_n$ as $u = \sum_{j=1}^n a_j v_j$. Also we have $w_j^* u = \sum_{j=1}^n a_j w_j^* v_j = a_j$. We thus have the below equation

$$u = \sum_{j=1}^n (w_j^* u) v_j \tag{14}$$

for any vector $u \in \mathbb{C}^n$. We know the below relation due to $v_k$ being the eigenvector of $A$ with eigen value $\lambda_k$

$$A v_k = \lambda_k v_k \tag{15}$$

We also can write the following in terms of the dual basis (since $A$ is a normal matrix)

$$
\begin{aligned}
w_k^* A &= \sum_j \lambda_j w_k^* v_j w_j^* \\
w_k^* A &= \lambda_k w_k^*
\end{aligned}
\tag{16}
$$

We now differentiate 21 using the product rule of differentiation to get

$$\dot{A}v_k + A\dot{v}_k = \dot{\lambda}_k v_k + \lambda_k \dot{v}_k \tag{17}$$

Taking the inner product of the equation 23 with $w_k^*$, and using 22 we obtain:

$$\dot{A}v_k + A\dot{v}_k = \dot{\lambda}_k v_k + \lambda_k \dot{v}_k$$
$$w_k^* \dot{A}v_k + w_k^* A\dot{v}_k = w_k^* \dot{\lambda}_k v_k + w_k^* \lambda_k \dot{v}_k$$
$$w_k^* \dot{A}v_k + \lambda_k w_k^* \dot{v}_k = \dot{\lambda}_k w_k^* v_k + \lambda_k w_k^* \dot{v}_k \tag{18}$$
$$w_k^* \dot{A}v_k = \dot{\lambda}_k$$
$$\dot{\lambda}_k = w_k^* \dot{A}v_k$$

Assuming $\lambda_k^0$ to be the eigenvalue at the start, we can get the value after time $t$ by simply integrating as follows,

$$\lambda_k^t = \lambda_k^0 + \int_0^t \omega_k \dot{A} v_{k_0} dt \tag{19}$$

Thus from the above result and equation 13 we have

$$\begin{aligned}
\left\| L_{\mathcal{J}_\mathcal{D}} y - (\mu_k + \lambda_l^0) y \right\| &= \left\| [a_k \boxtimes \{b_l^t\}] diag(\{\mu_k + \lambda_l^t - (\mu_k + \lambda_l^0)\}) \right\| \\
&= \left\| [a_k \boxtimes \{b_l^t\}] diag(\lambda_l^t - \lambda_l^0) \right\| \\
&= \left\| [a_k \boxtimes \{b_l^t\}] diag(\int_0^t \omega_k \dot{A} v_{k_0} dt) \right\| \\
&\leq \left\| [a_k \boxtimes \{b_l^t\}] \right\| \left\| diag(\int_0^t \omega_k \dot{A} v_{k_0} dt) \right\| \\
&\leq \left\| \int_0^T \int_0^t \omega_k \dot{A} v_{k_0} dt dt \right\| \\
&\leq \left\| \int_0^T \int_0^t \left\| \omega_k \dot{A} v_{k_0} \right\| dt dt \right\| \\
&\leq \left\| \int_0^T \int_0^t \left\| \dot{A} \right\| dt dt \right\| \\
&\leq \left\| \int_0^T \int_0^t \delta N dt dt \right\| \\
&\leq \left\| \int_0^T \delta N T dt \right\| \\
&\leq \delta N T^2 \\
&\leq \mathcal{O}(\delta) \\
\left\| L_{\mathcal{J}_\mathcal{D}} y - (\mu_k + \lambda_l^0) y \right\| &\leq \epsilon \\
\left\| L_{\mathcal{J}_\mathcal{D}} - (\mu_k + \lambda_l^0) \right\| &\leq \epsilon
\end{aligned}$$

Thus $\mu_k + \lambda_l^0$ is in the pseudospectrum of $L_{\mathcal{J}_\mathcal{D}}$ and so $y = [a_k \boxtimes \{b_l^t\}]$ is one of the solutions to the objective with the pseudospectrum relaxation. Thus it follows that $[\mathbf{\Psi}_T \boxtimes \{\mathbf{\Psi}_{G_t}\}]$ forms a basis of the solution to the defined objective, where $\mathbf{\Psi}_T$ and $\mathbf{\Psi}_{G_t}$ have $a_k^*$ and $b_l^t$ as their row spaces respectively.

$\square$

The above result gives us the definition of *EFT* in terms of the Kronecker product of the Time Fourier Transform and the Graph Fourier Transform of the graph at each time. While both *EFT* and *AD* are solutions to the pseudospectrum relaxed objectives they are not equal in general. To see this, we

first need to look at the eigenvectors of $L_{\mathcal{J}_{\mathcal{D}}}$. Let $\boldsymbol{\Psi}_{AD}$ be the matrix whose rows form the right eigenvectors of $L_{\mathcal{J}_{\mathcal{D}}}$. Below we state and prove the result of equivalence between $\boldsymbol{\Psi}_D$ and $\boldsymbol{\Psi}_{AD}$ for the general case of dynamic graphs using a counter example

**Remark 1.** *In general, the collective dynamic graph fourier transform as defined by the operator $\boldsymbol{\Psi}_D$ does not form the eigenspace of the spectrum of $L_{\mathcal{J}_{\mathcal{D}}}$ i.e. $\boldsymbol{\Psi}_D \neq \boldsymbol{\Psi}_{AD}$.*

*Proof.* It is sufficient to show a single counter example to conclude the statement.

Consider the below weighted adjacency matrix for a certain graph at time $t_0$

$$G_0 = \begin{bmatrix} 1 & 0.5 \\ 0.5 & 1 \end{bmatrix}$$

Let this change to the following in the next timestep $t_1$

$$G_1 = \begin{bmatrix} 1 & 0.6 \\ 0.6 & 1 \end{bmatrix}$$

The Laplacian $L_{\mathcal{J}_{\mathcal{D}}}$ is given by

$$L_{\mathcal{J}_{\mathcal{D}}} = \begin{bmatrix} 1.5 & -0.5 & -1. & -0. \\ -0.5 & 1.5 & -0. & -1. \\ -1. & -0. & 1.6 & -0.6 \\ -0. & -1. & -0.6 & 1.6 \end{bmatrix}$$

The EFT matrix $\boldsymbol{\Psi}_D$ is

$$\boldsymbol{\Psi}_D = \begin{bmatrix} 0.5 & -0.5 & -0.5 & 0.5 \\ 0.5 & 0.5 & -0.5 & -0.5 \\ 0.5 & -0.5 & 0.5 & -0.5 \\ 0.5 & 0.5 & 0.5 & 0.5 \end{bmatrix}$$

Similarly the matrix $\boldsymbol{\Psi}_{AD}$ comes out to be the following (upto sign and row wise permutations)

$$\boldsymbol{\Psi}_{AD} = \begin{bmatrix} 0.47 & -0.47 & -0.52 & 0.52 \\ 0.5 & 0.5 & -0.5 & -0.5 \\ 0.52 & -0.52 & 0.47 & -0.47 \\ 0.5 & 0.5 & 0.5 & 0.5 \end{bmatrix}$$

From the above we can see the two matrices differ and so we have a counter example.

$\square$

From the above result we can see that in the general case of dynamic graphs the defined *EFT* and the eigen decomposition of the defined Laplacian $L_{\mathcal{J}_{\mathcal{D}}}$ are not the same. Thus we can have an alternate definition of the collective dynamic graph fourier transform in terms of the decomposition of the joint Laplacian $L_{\mathcal{J}_{\mathcal{D}}}$. We term $\boldsymbol{\Psi}_{AD}$ as the *Absolute Drcomposition* or *AD* for brevity.

Both *EFT* and *AD* have their own advantages. *EFT* has a simple primal definition and is easy to compute whereas *AD* has a beautiful physical interpretation. Even though *EFT* and *AD* are not exactly the same, in order to have desirable properties of both we can define approximation bounds that inform under what conditions the two transforms can be used interchangeably upto the approximation error. We work under the below assumptions for weighted graphs in order to bound the two transforms:

1. The rate of change of the graph with time is bounded
2. The eigenvalues of the graph Laplacian at any given timestep and between timesteps has a multiplicity of 1

The condition 2 is required for stability of the bound and can be enforced for example by adding random perturbations to the matrix.

Based on these assumptions we state and prove the bounds between *EFT* and *AD* below

**Theorem 1.** *Considering bounded changes in a graph $G$ with $N$ nodes over time $T$, the norm of the difference between* EFT *($\mathbf{\Psi}_D$) and* AD *($\mathbf{\Psi}_{AD}$) is bounded as follows: $\|\mathbf{\Psi}_D - \mathbf{\Psi}_{AD}\| \le \mathcal{O}\left(\frac{N^{\frac{3}{2}}T\omega_{max}^{\frac{1}{2}}}{(\Delta\lambda_G)_{min}} + \frac{N^{\frac{3}{2}}T\omega_{max}^2}{(\Delta\lambda_J)_{min}}\right)\left(\left\|\dot{L}_G\right\|\right)_{max}$ where $(\Delta\lambda_G)_{min}$ and $(\Delta\lambda_J)_{min}$ refer to the minimum difference between the eigenvalues of matrices $L_G$ and $L_{\mathcal{J}_D}$ respectively, $\dot{L}_G$ is the rate of change of $L_G$ and $\omega_{max} = 2\pi$.*

*Proof.* Consider $L_J$ to be the Laplacian of collective graph when the graphs are static with time. We can show this, in a similar manner as $L_{\mathcal{J}_D}$ to be $L_J = L_T \otimes I_N + I_T \otimes L_G$, where $L_G$ is the Laplacian of the static graph. Let $\mathbf{\Psi}_J$ be the matrix whose rows form the left eigenvectors of $L_J$. Now we consider the graph to change infinitesimally with $L_G$ as the starting state of the Laplacian. We intend to bound the frobenius norm $\|\mathbf{\Psi}_D - \mathbf{\Psi}_{AD}\|$. We can manipulate this as follows

$$\|\mathbf{\Psi}_D - \mathbf{\Psi}_{AD}\| = \|\mathbf{\Psi}_D - \mathbf{\Psi}_{AD} + \mathbf{\Psi}_J - \mathbf{\Psi}_J\|$$
$$= \|\mathbf{\Psi}_D - \mathbf{\Psi}_J + \mathbf{\Psi}_J - \mathbf{\Psi}_{AD}\|$$
$$\le \|\mathbf{\Psi}_D - \mathbf{\Psi}_J\| + \|\mathbf{\Psi}_J - \mathbf{\Psi}_{AD}\|$$

We thus find the bound in two parts first for the error between $\mathbf{\Psi}_D, \mathbf{\Psi}_J$ and second for the error between $\mathbf{\Psi}_J, \mathbf{\Psi}_{AD}$.

In order to bound the matrices (which are formed by the eigenvectors) we first attempt to bound the vectors forming the matrix. For this we study the rate of change of the vectors with time (as the graph evolves) using the language of calculus. For deeper insights into this and the stability of eigenvectors/values we refer the interested reader to (Tao, 2008). Consider a normal matrix $A$ of which the eigenvectors $v_1, \ldots, v_n$ form a basis of $\mathbb{C}^n$. Also we consider $w_1, \ldots, w_n$ be the dual basis, i.e. $w_j^* v_k = \delta_{jk}$ for all $1 \le j, k \le n$, where $\delta_{jk}$ is the Kronecker delta and

$$\delta_{jk} = \begin{cases} 1, & \text{if } j = k \\ 0, & \text{otherwise} \end{cases}$$

Since the eigenvectors form a basis we can represent any vector $u$ as a linear combination of $v_1, \ldots, v_n$ as $u = \sum_{j=1}^n a_j v_j$. Also we have $w_j^* u = \sum_{j=1}^n a_j w_j^* v_j = a_j$. We thus have the below equation

$$u = \sum_{j=1}^n (w_j^* u) v_j \tag{20}$$

for any vector $u \in \mathbb{C}^n$. We know the below relation due to $v_k$ being the eigenvector of $A$ with eigen value $\lambda_k$

$$A v_k = \lambda_k v_k \tag{21}$$

We also can write the following in terms of the dual basis (since $A$ is a normal matrix)

$$w_k^* A = \sum_j \lambda_j w_k^* v_j w_j^*$$
$$w_k^* A = \lambda_k w_k^* \tag{22}$$

We now differentiate 21 using the product rule of differentiation to get

$$\dot{A} v_k + A \dot{v}_k = \dot{\lambda}_k v_k + \lambda_k \dot{v}_k \tag{23}$$

Taking the inner product of the equation 23 with $w_k^*$, and using 22 we obtain:

$$\dot{A} v_k + A \dot{v}_k = \dot{\lambda}_k v_k + \lambda_k \dot{v}_k$$
$$w_k^* \dot{A} v_k + w_k^* A \dot{v}_k = w_k^* \dot{\lambda}_k v_k + w_k^* \lambda_k \dot{v}_k$$
$$w_k^* \dot{A} v_k + \lambda_k w_k^* \dot{v}_k = \dot{\lambda}_k w_k^* v_k + \lambda_k w_k^* \dot{v}_k \tag{24}$$
$$w_k^* \dot{A} v_k = \dot{\lambda}_k$$
$$\dot{\lambda}_k = w_k^* \dot{A} v_k$$

In our case since A is normal, we have the eigenbasis $v_k$ as an orthonormal set, and the dual basis $w_k$ is identical to $v_k$.

We are interested in how the eigenvectors change with time. Taking the inner product of equation 23 with $w_j^*$ for $j \neq k$, we get

$$
\begin{aligned}
\dot{A}v_k + A\dot{v}_k &= \dot{\lambda}_k v_k + \lambda_k \dot{v}_k \\
w_j^* \dot{A}v_k + w_j^* A\dot{v}_k &= w_j^* \dot{\lambda}_k v_k + w_j^* \lambda_k \dot{v}_k \\
w_j^* \dot{A}v_k + \lambda_j w_j^* \dot{v}_k &= \dot{\lambda}_k w_j^* v_k + w_j^* \lambda_k \dot{v}_k \\
w_j^* \dot{A}v_k + \lambda_j w_j^* \dot{v}_k &= w_j^* \lambda_k \dot{v}_k \\
w_j^* \dot{A}v_k + \lambda_j w_j^* \dot{v}_k - w_j^* \lambda_k \dot{v}_k &= 0 \\
w_j^* \dot{A}v_k + (\lambda_j - \lambda_k) w_j^* \dot{v}_k &= 0 \\
w_j^* \dot{v}_k &= \frac{w_j^* \dot{A}v_k}{(\lambda_k - \lambda_j)}
\end{aligned}
\tag{25}
$$

Using the above in 20 we obtain the following

$$
\begin{aligned}
\dot{v}_k &= \sum_{j=1}^{n} (w_j^* \dot{v}_k) v_j \\
\dot{v}_k &= \sum_{j \neq k} (w_j^* \dot{v}_k) v_j + (w_k^* \dot{v}_k) v_k \\
\dot{v}_k &= \sum_{j \neq k} \frac{w_j^* \dot{A}v_k}{\lambda_k - \lambda_j} v_j + (w_k^* \dot{v}_k) v_k
\end{aligned}
\tag{26}
$$

We consider the change in $A$ so that the resulting matrix is also normal. Thus the eigenvectors of the resulting matrix will also be orthonormal. This imples all the vectors lie on the surface of the unit sphere in $\mathbb{C}^n$ and so the change in the eigenvectors should be along the surface of this sphere. As such $\dot{A}v_k$ will be tangential to the sphere at $v_k$ and so $\dot{v}_k^\top v_k = 0$. Note this need not be the case in general if we consider non-unit vectors (that could also be eigenvectors). Thus we can represent $\dot{v}_k = 0v_k + \sum_{j \neq k} b_j v_j$. Thus we have

$$
\begin{aligned}
\dot{v}_k &= v_k + \sum_{j \neq k} b_j v_j \\
w_k^* \dot{v}_k &= w_k^* \left( \sum_{j \neq k} b_j v_j \right) \\
w_k^* \dot{v}_k &= \left( \sum_{j \neq k} b_j w_k^* v_j \right) \\
w_k^* \dot{v}_k &= 0
\end{aligned}
$$

Using the above equations and the consideration that $\|v_j\| = 1$ we have

$$\dot{v}_k = \sum_{j \neq k} \frac{w_j^* \dot{A} v_k}{\lambda_k - \lambda_j} v_j$$

$$\|\dot{v}_k\| = \left\| \sum_{j \neq k} \frac{w_j^* \dot{A} v_k}{\lambda_k - \lambda_j} v_j \right\|$$

$$\leq \sum_{j \neq k} \left\| \frac{w_j^* \dot{A} v_k}{\lambda_k - \lambda_j} v_j \right\|$$

$$\leq \sum_{j \neq k} \left\| \frac{w_j^* \dot{A} v_k}{\lambda_k - \lambda_j} \right\| \|v_j\|$$

Since we consider orthonormal vectors $\|v_j\| = 1$

$$\therefore \|\dot{v}_k\| \leq \sum_{j \neq k} \left\| \frac{w_j^* \dot{A} v_k}{\lambda_k - \lambda_j} \right\|$$

$$\leq \sum_{j \neq k} \frac{\left\| w_j^* \dot{A} v_k \right\|}{\|\lambda_k - \lambda_j\|}$$

$$\leq \sum_{j \neq k} \frac{\sigma(\dot{A})}{\|\lambda_k - \lambda_j\|}$$

$$\leq \sum_{j \neq k} \frac{\left\| \dot{A} \right\|}{\|\lambda_k - \lambda_j\|}$$

$$\leq \sum_{j \neq k} \frac{\left\| \dot{A} \right\|}{(\Delta\lambda)_{min}}$$

$$\leq \frac{N-1}{(\Delta\lambda)_{min}} \left\| \dot{A} \right\|$$

where $\sigma(.)$ is the operator norm and $(\Delta\lambda)_{min}$ is the absolute of the minimum difference between the eigenvalues of $A$. in the above we have seen how the change in eigenvectors is bounded by the change in the matrix. Using this result we now attempt to bound the change in the required transform matrices.

For the first part we bound $\|\mathbf{\Psi}_D - \mathbf{\Psi}_J\|$. Let $\Delta v$ represent the (infinitesimal) change in the eigenvectors of $L_G$ in time $t$ and let $\Delta v_i$ be the infinitesimal change per unit time in the vector at step $i$. Thus using the triangle inequality we have the below equations

$$\|\Delta v\| \leq \sum_i \|\Delta v_i\|$$

$$\|\Delta v\| \leq \int_t \|\dot{v}(t)\| dt$$

Using the above derived inequality $\|\dot{v}_k\| \leq \frac{N-1}{(\Delta\lambda)_{min}} \left\|\dot{A}\right\|$ for $L_G$ we have

$$\|\Delta v\| \leq \int_t \left\| \frac{N-1}{(\Delta\lambda)_{min}} \left\|\dot{L}_G(t)\right\| \right\| dt$$

$$\leq \int_t \frac{N-1}{(\Delta\lambda)_{min}} \left(\left\|\dot{L}_G(t)\right\|\right)_{max} dt$$

Finally taking the maximum norm of the rate of change in $L_G$ over the entire time duration we have the following

$$\|\Delta v\| \leq \frac{N-1}{(\Delta\lambda)_{min}} \left(\left\|\dot{L}_G\right\|\right)_{max} \int_t dt$$

$$\|\Delta v\| \leq \frac{N-1}{(\Delta\lambda)_{min}} \left(\left\|\dot{L}_G\right\|\right)_{max} t$$

$$\leq \frac{\mathcal{O}(N-1)T}{(\Delta\lambda)_{min}} \left(\left\|\dot{L}_G\right\|\right)_{max}$$

where $\left(\dot{L}_G\right)_{max}$ is the maximum of the norm of the rate of change of $L_G$ over all timesteps considered and we absorb the total time $t$ into the constant factor considering finite time. Thus we have,

$$\left\|\dot{\boldsymbol{\Psi}}_G\right\| = \sqrt{\sum_i \|v_i\|^2}$$

$$\leq \frac{\sqrt{N}(N-1)T}{(\Delta\lambda_G)_{min}} \left(\dot{L}_G\right)_{max}$$

$$\leq \frac{\mathcal{O}(N^{\frac{3}{2}}T)}{(\Delta\lambda_G)_{min}} \left(\dot{L}_G\right)_{max}$$

Thus using theorem 8 from (Lancaster & Farahat, 1972) and the fact that $\|\boldsymbol{\Psi}_T^t\| = 1$ we have,

$$\|\boldsymbol{\Psi}_D - \boldsymbol{\Psi}_J\| = \|([\boldsymbol{\Psi}_T \boxtimes (\{\boldsymbol{\Psi}_{G_t}\} - \boldsymbol{\Psi}_G)])\|$$

$$= \left(\int_\omega \|\boldsymbol{\Psi}_T^\omega \otimes (\boldsymbol{\Psi}_{G_\omega} - \boldsymbol{\Psi}_G)\|^2\right)^{\frac{1}{2}}$$

$$\leq \left(\int_\omega \|\boldsymbol{\Psi}_T^\omega\|^2 \|(\boldsymbol{\Psi}_{G_\omega} - \boldsymbol{\Psi}_G)\|^2\right)^{\frac{1}{2}}$$

$$\leq \left(\int_\omega \|(\boldsymbol{\Psi}_{G_\omega} - \boldsymbol{\Psi}_G)\|^2\right)^{\frac{1}{2}}$$

$$\leq \left(\int_\omega \|\Delta\boldsymbol{\Psi}_{G_\omega}\|^2\right)^{\frac{1}{2}}$$

$$\leq \left(\int_\omega \left(\frac{\mathcal{O}(N^{\frac{3}{2}}T)}{(\Delta\lambda_G)_{min}} \left\|\dot{L}_{G_\omega}\right\|_{max}\right)^2\right)^{\frac{1}{2}}$$

$$\leq \left(\omega_{max}\left(\frac{\mathcal{O}(N^{\frac{3}{2}}T)}{(\Delta\lambda_G)_{min}} \left\|\dot{L}_G\right\|_{max}\right)^2\right)^{\frac{1}{2}}$$

$$\|\boldsymbol{\Psi}_D - \boldsymbol{\Psi}_J\| \leq \left(\frac{\mathcal{O}(N^{\frac{3}{2}}T\omega_{max}^{\frac{1}{2}})}{(\Delta\lambda_G)_{min}} \left\|\dot{L}_G\right\|_{max}\right)$$

where $T$ is the time duration over which the evolution ghof the graphs is considered.

For the second part we can show that

$$\|\mathbf{\Psi}_J - \mathbf{\Psi}_{AD}\| = \left\|\dot{\mathbf{\Psi}}_J\right\|$$
$$= \sqrt{\int_{i=0}^{N\omega_{max}} \|v_i\|^2}$$
$$\leq \frac{\sqrt{N\omega_{max}}(N\omega_{max}-1)T}{(\Delta\lambda_J)_{min}} \left(\dot{L}_J\right)_{max}$$
$$\leq \frac{\mathcal{O}((N\omega_{max})^{\frac{3}{2}}T)}{(\Delta\lambda_J)_{min}} \left(\dot{L}_J\right)_{max}$$

Also we have,

$$L_J = L_T \oplus L_G$$
$$= L_T \otimes I_N + I_T \otimes L_G$$
$$\dot{L}_J = I_T \otimes \dot{L}_G$$
$$\left\|\dot{L}_J\right\| = \left\|I_T \otimes \dot{L}_G\right\|$$
$$\left\|\dot{L}_J\right\| = \|I_T\|\left\|\dot{L}_G\right\|$$
$$\left\|\dot{L}_J\right\| = \int_\omega (d\omega)^{\frac{1}{2}} \left\|\dot{L}_G\right\|$$
$$\left\|\dot{L}_J\right\| = \sqrt{\omega}\left\|\dot{L}_G\right\|$$

$$\therefore \|\mathbf{\Psi}_J - \mathbf{\Psi}_{AD}\| \leq \frac{\mathcal{O}(N^{\frac{3}{2}})}{(\Delta\lambda_J)_{min}} \left(\dot{L}_J\right)_{max}$$
$$\leq \frac{\mathcal{O}(N^{\frac{3}{2}}\omega^2 T)}{(\Delta\lambda_J)_{min}} \left(\dot{L}_G\right)_{max}$$

Combining the two parts we get the result

$$\|\mathbf{\Psi}_D - \mathbf{\Psi}_{AD}\| \leq \|\mathbf{\Psi}_D - \mathbf{\Psi}_J\| + \|\mathbf{\Psi}_J - \mathbf{\Psi}_{AD}\|$$
$$\leq \mathcal{O}\left(\frac{N^{\frac{3}{2}}T\omega_{max}^{\frac{1}{2}}}{(\Delta\lambda_G)_{min}} + \frac{N^{\frac{3}{2}}T\omega_{max}^2}{(\Delta\lambda_J)_{min}}\right)\left(\left\|\dot{L}_G\right\|\right)_{max} \tag{27}$$

For the discrete case this bound becomes

$$\|\mathbf{\Psi}_D - \mathbf{\Psi}_{AD}\| \leq \mathcal{O}\left(\frac{(NT)^{\frac{3}{2}}}{(\Delta\lambda_G)_{min}} + \frac{(NT^2)^{\frac{3}{2}}}{(\Delta\lambda_J)_{min}}\right)\left(\left\|\dot{L}_G\right\|\right)_{max} \tag{28}$$

$\square$

We thus see that as the graph evolves infinitesimally the difference between $\mathbf{\Psi}_D$ and $\mathbf{\Psi}_{AD}$ is bounded from above by the change in the graph matrix representation. This is desirable since it allows us to approximate $\mathbf{\Psi}_{AD}$ (formed by the eigendecomposition of $L_{\mathcal{J}_D}$) which has a physical interpretation using the defined $\mathbf{\Psi}_D$ which is simple to compute, when the graph changes in a stable manner. In such cases, *EFT* therefore characterizes signals on the dynamic graph by its proximity (projection) to the optimizers of $S_2(X)$ meaning high (collective dynamic graph) frequency components correspond to sharply varying signals and low frequency components to smoother signals. Having derived the transform, we next state and prove the properties of the proposed transform in the next section.

## C  Properties of Proposed Transform

Having designed the Evolving Graph Fourier Transform, we now look at some of the properties of the transform. The below defines some properties of *EFT* before learning its representations and applying to downstream tasks (proofs are in appendix section B).

---

**Property 1.** *(Equivalence in special case) Consider $\mathbf{\Psi}_T$ to be the time Fourier transform and $\mathbf{\Psi}_{G_t}$ to be the Graph Fourier transform at time t. Let $\mathbf{\Psi}_{\mathcal{J}_\mathcal{D}}$ be the Graph fourier transform of $\mathcal{J}_\mathcal{D}$. In the special case of $G_{t_i} = G_{t_j} \forall i, j \in \{T\}$ we have $(\mathbf{\Psi}_{\mathcal{J}_\mathcal{D}})_i^j = (\mathbf{\Psi}_D)_i^j = (\mathbf{\Psi}_T \otimes \{\mathbf{\Psi}_{G_t}\})_i^{j\left\lfloor\frac{j}{N}\right\rfloor}$.*

**Property 2.** *EFT is an invertible transform and the inverse is given by $\mathbf{EFT}^{-1}(\hat{X})_i^j = \left(\mathbf{\Psi}_G^{-1}\hat{X}\right)_i^{kk}\left(\mathbf{\Psi}_T^{\top *}\right)_k^j$ in matrix form and $\mathbf{EFT}^{-1}(\hat{x})_{j*N+i} = \left(\mathbf{\Psi}_T^* \otimes \mathbf{\Psi}_G^{-1}\right)_{j*N+i}^{k\left\lfloor\frac{k}{N}\right\rfloor} \hat{x}_k$ in vector form.*

**Property 3.** *EFT is a unitary transform if and only if GFT is unitary at all timesteps considered i.e. $\mathbf{\Psi}_D\mathbf{\Psi}_D^* = I_{NT}$ iff $\mathbf{\Psi}_{G_t}\mathbf{\Psi}_{G_t}^* = I_N, \forall t$.*

**Property 4.** *EFT is invariant to the order of application of DFT or GFT on signal X.*

---

Property 3 allows us to define the stability of the proposed transform. Consider the EFT matrix $E$ and the signal vector $x$ (normalized). The transform would be given by $Ex$. Now consider the perturbed matrix $E + \epsilon$, where $\epsilon$ is the (fixed) perturbation. The relative difference between the output would be $\|(E + \epsilon)x - Ex\|/\|Ex\| = \|\epsilon x\|/\|Ex\|$. Since $E$ is orthogonal, $x$ is not in the null space of $E$ and so the relative difference is bounded by $\epsilon$. So a small change in $E$ should cause a small change in the output as desired.

As seen in property 1, *EFT* can be simulated by *GFT* in the special case that the graph structure does not change with time. The illustration between other transforms is in Figure 1. The figure shows transforms (*GFT, JFT, DFT, EFT*) in a circle, and arrows from one transform to the next indicate that the source transform can be obtained by the destination transform using the simulation annotated on the edges. Please note that the analysis has been performed for one-dimensional signals. However, the same holds true for higher dimensions as well by conducting the *EFT* dimension-wise. Here dimension-wise means the feature dimension of a node. Each node may have a multidimensional signal residing on it and the EFT can be independently applied to each channel or dimension of the node signals on the dynamic graph. Below subsection provides proofs for the above stated properties.

### C.1  Proofs of Properties

In this section we now prove the properties stated above. We repeat the statements for completeness. Though *EFT* and *AD* are not same in the general case they are equivalent when the graph structure does not change with time. Below result proves the result for the discrete case with graphs sampled at uniform timesteps

**Property 5.** *(Special Equivalence between AD and EFT) Consider $\mathbf{\Psi}_T$ to be the time fourier transform and $\mathbf{\Psi}_{G_t}$ to be the Graph fourier transform at time t. Let $\mathbf{\Psi}_{\mathcal{J}_\mathcal{D}}$ be the Graph fourier transform of $\mathcal{J}_\mathcal{D}$. In the special case of $G_{t_i} = G_{t_j} \forall i, j \in \{T\}$ we have $(\mathbf{\Psi}_{\mathcal{J}_\mathcal{D}})_i^j = (\mathbf{\Psi}_D)_i^j = (\mathbf{\Psi}_T \otimes \{\mathbf{\Psi}_{G_t}\})_i^{j\left\lfloor\frac{j}{N}\right\rfloor}$.*

*Proof.* As before consider $\{L_{G_t}\} \in R^{N \times N \times T}$ to be the Laplacian of the graphs at each timestep with eigenvalues $\lambda_i^t$ where $i \in N, t \in T$. Let $L_T \in R^{T \times T}$ be the Laplacian of the time adjacency matrix with eigenvalues $\mu_j$ where $j \in T$. The Laplacian of the collective graph $\mathcal{J}_\mathcal{D}$ is expressed as

$$(L_{\mathcal{J}_\mathcal{D}})_i^j = (L_T \oplus \{L_{G_t}\})_i^{j\left\lfloor\frac{j}{N}\right\rfloor} = L_T \otimes I_N + (I_T \otimes \{L_{G_t}\})_i^{j\left\lfloor\frac{j}{N}\right\rfloor}$$

Consider $x_1, x_2, \ldots x_p$ to be the linearly independent right eigenvectors of $L_T$ and $z_1^t, z_2^t, \ldots z_{q_t}^t$ to be the linearly independent right eigenvectors of $L_{G_t}$. Consider the vector $y_j = (x_k \otimes z_l^t)_j^{\left\lfloor\frac{j}{N}\right\rfloor}, y \in R^{NT}$.

Then we have

$$
\begin{aligned}
(L_{\mathcal{J}_\mathcal{D}}y)_i &= (L_T \otimes I_N)_i^j y_j + (I_T \otimes \{L_{G_t}\})_i^{j\lfloor\frac{j}{N}\rfloor} y_j \\
&= (L_T \otimes \{I_N\})_i^{j\lfloor\frac{j}{N}\rfloor}(x_k \otimes z_l^t)_j^{\lfloor\frac{j}{N}\rfloor} + (I_T \otimes \{L_{G_t}\})_i^{j\lfloor\frac{j}{N}\rfloor}(x_k \otimes z_l^t)_j^{\lfloor\frac{j}{N}\rfloor} \\
&= (L_T \otimes \{I_N\}\Box x_k \otimes z_l^t)_i^{\lfloor\frac{i}{N}\rfloor} + (I_T \otimes \{L_{G_t}\Box x_k \otimes z_l^t\})_i^{\lfloor\frac{i}{N}\rfloor} \\
&= (L_T x_k \otimes \{I_N\}\Box z_l^t)_i^{\lfloor\frac{i}{N}\rfloor} + (I_T x_k \otimes \{L_{G_t}\Box z_l^t\})_i^{\lfloor\frac{i}{N}\rfloor} \\
&= (\mu_k x_k \otimes z_l^t)_i^{\lfloor\frac{i}{N}\rfloor} + (x_k \otimes \{\lambda_l^t z_l^t\})_i^{\lfloor\frac{i}{N}\rfloor} \\
&= (\mu_k x_k \otimes z_l^t + x_k \otimes \{\lambda_l^t z_l^t\})_i^{\lfloor\frac{i}{N}\rfloor} \\
&= (x_k \otimes z_l^t diag(\{\mu_k\}) + x_k \otimes \{z_l^t\}diag(\{\lambda_l^t\}))_i^{\lfloor\frac{i}{N}\rfloor} \\
&= ((x_k \otimes z_l^t)diag(\{\mu_k + \lambda_l^t\}))_i^{\lfloor\frac{i}{N}\rfloor}
\end{aligned}
$$

where $\Box$ indicates timestep (column) wise product and $diag(.)$ operator converts a vector to a diagonal matrix. In the special case where $G_{ti} = G_{tj}\forall i, j \in T$ we have $\lambda_l^{ti} = \lambda_l^{tj}$. Thus we get

$$
\begin{aligned}
(L_{\mathcal{J}_\mathcal{D}}y)_i &= ((x_k \otimes z_l^t)diag(\{\mu_k + \lambda_l I_T\}))_i^{\lfloor\frac{i}{N}\rfloor} \\
&= (\mu_k + \lambda_l(x_k \otimes z_l^t)diag(\{I_T\}))_i^{\lfloor\frac{i}{N}\rfloor} \\
&= (\mu_k + \lambda_l(x_k \otimes z_l^t))_i^{\lfloor\frac{i}{N}\rfloor} \\
&= (\mu_k + \lambda_l)y_i
\end{aligned}
$$

Thus $y_j = (x_k \otimes z_l^t)_j^{\lfloor\frac{j}{N}\rfloor}$ is the eigenvector of $L_{\mathbf{J_D}}$ with eigenvalue $\mu_k + \lambda_l$. But $y$ is nothing but one of the columns of $\boldsymbol{\Psi}_D^*$. By the rank nullity theorem, the row spaces of the transform matrices $\boldsymbol{\Psi}_D$ and $GFT$ of $\mathcal{J}_\mathcal{D}$ share the same orthogonal basis. Thus the two transforms are equivalent in this case. $\square$

Note the eigenvalues $(\Lambda_T \oplus \Lambda_G)$ obtained in the result above are exactly the ones used for plotting the frequency response of *EFT* as we compress the sequence of graphs into a single dynamic graph.

Next we prove some properties of *EFT* as stated in the main paper

**Property 6.** *EFT is an invertible transform and the inverse is given by* $\mathbf{EFT}^{-1}(\hat{X})_i^j = \left(\boldsymbol{\Psi}_G^{-1}\hat{X}\right)_i^{kk}\left(\boldsymbol{\Psi}_T^{\top *}\right)_k^j$ *in matrix form and* $\mathbf{EFT}^{-1}(\hat{x})_{j*N+i} = \left(\boldsymbol{\Psi}_T^* \otimes \boldsymbol{\Psi}_G^{-1}\right)_{j*N+i}^{k\lfloor\frac{k}{N}\rfloor}\hat{x}_k$ *in vector form.*

*Proof.* We begin by noting the expression for *EFT* ($\boldsymbol{\Psi}_D$)

$$
(\boldsymbol{\Psi}_D)_j^i = (\boldsymbol{\Psi}_T \otimes \boldsymbol{\Psi}_{G_t})_i^{j\lfloor\frac{j}{N}\rfloor}
$$

where $\boldsymbol{\Psi}_{G_t} \in R^{N\times N}$ is the graph fourier transform of the graph at time $t$, $\boldsymbol{\Psi}_T \in R^{T\times T}$ is the time fourier transform. Let $\boldsymbol{\Phi}_{G_t} = \boldsymbol{\Psi}_{G_t}^{-1}$ be the inverse graph fourier transform of the graph at timestep $t$ and $\boldsymbol{\Phi}_T = \boldsymbol{\Psi}_T^*$ be the inverse time fourier transform.

We can write $\boldsymbol{\Psi}_D$ as a block matrix in the following form

$$
\begin{aligned}
\boldsymbol{\Psi}_D &= \begin{bmatrix} CB^1, CB^2, \dots CB^T \end{bmatrix} \\
CB^i &= \boldsymbol{\Psi}_T^i \otimes \boldsymbol{\Psi}_{G_i}
\end{aligned}
$$

where $\boldsymbol{\Psi}_T^i$ is the $i$-th column of $\boldsymbol{\Psi}_T$ and $CB^i \in R^{NT\times N}$.

Consider $\boldsymbol{\Phi}_D$ in a similar but row block format as follows

$$\boldsymbol{\Phi}_D = \begin{bmatrix} RB_1 \\ RB_2 \\ \vdots \\ RB_T \end{bmatrix} \tag{29}$$

$$RB_i = \boldsymbol{\Phi}_{Ti} \otimes \boldsymbol{\Phi}_{G_i} \tag{30}$$

where $\boldsymbol{\Phi}_{Ti}$ is the $i$-th row of $\boldsymbol{\Phi}_T$ and $RB_i \in R^{N \times NT}$.

Now taking the matrix product of $\boldsymbol{\Phi}_D$ and $\boldsymbol{\Psi}_D$ we get

$$\begin{aligned}
\boldsymbol{\Phi}_D \boldsymbol{\Psi}_D &= \begin{bmatrix} RB_1 \\ RB_2 \\ \vdots \\ RB_T \end{bmatrix} \begin{bmatrix} CB^1 & CB^2 & \dots CB^T \end{bmatrix} \\
&= \begin{bmatrix} RB_1 CB^1 & RB_1 CB^2 \dots \\ RB_2 CB^1 & RB_2 CB^2 \dots \\ \vdots \\ RB_T CB^1 & RB_T CB^2 \dots \end{bmatrix}
\end{aligned}$$

We can verify that $RB_i CB^j$ evaluates to the following

$$RB_i CB^j = (\boldsymbol{\Phi}_{Ti} \otimes \boldsymbol{\Phi}_{G_i}) \left( \boldsymbol{\Psi}_T^j \boldsymbol{\Psi}_{G_j} \right) \tag{31}$$

$$= \left( \boldsymbol{\Phi}_{Ti} \boldsymbol{\Psi}_T^j \right) \otimes \left( \boldsymbol{\Phi}_{G_i} \boldsymbol{\Psi}_{G_j} \right) \tag{32}$$

$$\tag{33}$$

Now the columns of $\boldsymbol{\Phi}_T$ form the eigenvectors of a circulant matrix ($L_T$). Also we know that if columns form basis of column space then rows form the basis of the row space. Thus we have

$$\boldsymbol{\Phi}_{Ti} \boldsymbol{\Psi}_T^j = \begin{cases} 1, & \text{if } i = j \\ 0, & \text{otherwise} \end{cases} \tag{34}$$

$$\boldsymbol{\Phi}_{G_i} \boldsymbol{\Psi}_{G_i} = I_N \tag{35}$$

$$\therefore RB_i CB^j = \begin{cases} I_N, & \text{if } i = j \\ 0, & \text{otherwise} \end{cases} \tag{36}$$

$$\tag{37}$$

Thus we have shown that $\boldsymbol{\Phi}_D \boldsymbol{\Psi}_D = I_{NT}$. Thus $\boldsymbol{\Phi}_D$ is a left inverse of $\boldsymbol{\Psi}_D$. We know that for a square matrix left inverse is also the right inverse and can be readily verified in a similar manner. Thus *EFT* is invertible and the inverse of the transformed signal in vector form is $(\boldsymbol{\Phi}_T \otimes \{\boldsymbol{\Phi}_{G_t}\})_i^{j\lfloor \frac{j}{N} \rfloor} \hat{x}_j = \left(\boldsymbol{\Psi}_T^* \otimes \{\boldsymbol{\Psi}_{G_t}^{-1}\}\right)_i^{j\lfloor \frac{j}{N} \rfloor} \hat{x}_j$. Similarly for the matrix form of the signal we have the inverse of the transform given as $\left(\{\boldsymbol{\Phi}_{G_t}\}\hat{X}\right)_i^{jj} \left(\boldsymbol{\Phi}_T^\top\right)_j^k = \left(\{\boldsymbol{\Psi}_{G_t}^{-1}\}\hat{X}\right)_i^{jj} \left(\boldsymbol{\Psi}_T^{\top *}\right)_j^k$. $\qquad \square$

**Property 7.** *EFT is a unitary transform if and only if GFT is unitary at all timesteps considered i.e.* $\boldsymbol{\Psi}_D \boldsymbol{\Psi}_D^* = I_{NT}$ iff $\boldsymbol{\Psi}_{G_t} \boldsymbol{\Psi}_{G_t}^* = I_N, \forall t$

*Proof.* This property can be proved in a similar manner as in proof of property 6. The only difference here is we consider $\boldsymbol{\Phi}_D$ to be the transposed conjugate of $\boldsymbol{\Psi}_D$ rather than inverse i.e. $\boldsymbol{\Phi}_D = \boldsymbol{\Psi}_D^*$ and

also $\boldsymbol{\Phi}_{G_i} = \boldsymbol{\Psi}_{G_i}^*$. Similar to the previous proof we have the following

$$
\boldsymbol{\Phi}_D \boldsymbol{\Psi}_D = \begin{bmatrix} RB_1 \\ RB_2 \\ \vdots \\ RB_T \end{bmatrix} \begin{bmatrix} CB^1 & CB^2 & \ldots CB^T \end{bmatrix}
$$

$$
= \begin{bmatrix} RB_1 CB^1 & RB_1 CB^2 \ldots \\ RB_2 CB^1 & RB_2 CB^2 \ldots \\ \vdots & \\ RB_T CB^1 & RB_T CB^2 \ldots \end{bmatrix}
$$

$$
RB_i CB^j = (\boldsymbol{\Phi}_{Ti} \otimes \boldsymbol{\Phi}_{G_i}) \left( \boldsymbol{\Psi}_T^j \boldsymbol{\Psi}_{G_j} \right)
$$

$$
= \left( \boldsymbol{\Phi}_{Ti} \boldsymbol{\Psi}_T^j \right) \otimes \left( \boldsymbol{\Phi}_{G_i} \boldsymbol{\Psi}_{G_j} \right)
$$

$$
\therefore \boldsymbol{\Phi}_D \boldsymbol{\Psi}_D = \begin{bmatrix} \left( \boldsymbol{\Phi}_{T1} \boldsymbol{\Psi}_T^1 \right) \otimes \left( \boldsymbol{\Phi}_{G_1} \boldsymbol{\Psi}_{G_1} \right) & \left( \boldsymbol{\Phi}_{T1} \boldsymbol{\Psi}_T^2 \right) \otimes \left( \boldsymbol{\Phi}_{G_1} \boldsymbol{\Psi}_{G_2} \right) \ldots \\ \left( \boldsymbol{\Phi}_{T2} \boldsymbol{\Psi}_T^1 \right) \otimes \left( \boldsymbol{\Phi}_{G_2} \boldsymbol{\Psi}_{G_1} \right) & \left( \boldsymbol{\Phi}_{T2} \boldsymbol{\Psi}_T^2 \right) \otimes \left( \boldsymbol{\Phi}_{G_2} \boldsymbol{\Psi}_{G_2} \right) \ldots \\ \vdots & \\ \left( \boldsymbol{\Phi}_{TT} \boldsymbol{\Psi}_T^1 \right) \otimes \left( \boldsymbol{\Phi}_{G_T} \boldsymbol{\Psi}_{G_1} \right) & \left( \boldsymbol{\Phi}_{TT} \boldsymbol{\Psi}_T^2 \right) \otimes \left( \boldsymbol{\Phi}_{G_T} \boldsymbol{\Psi}_{G_2} \right) \ldots \end{bmatrix}
$$

$$
= \begin{bmatrix} 1 \otimes \left( \boldsymbol{\Phi}_{G_1} \boldsymbol{\Psi}_{G_1} \right) & 0 \otimes \left( \boldsymbol{\Phi}_{G_1} \boldsymbol{\Psi}_{G_2} \right) \ldots \\ 0 \otimes \left( \boldsymbol{\Phi}_{G_2} \boldsymbol{\Psi}_{G_1} \right) & 1 \otimes \left( \boldsymbol{\Phi}_{G_2} \boldsymbol{\Psi}_{G_2} \right) \ldots \\ \vdots & \\ 0 \otimes \left( \boldsymbol{\Phi}_{G_T} \boldsymbol{\Psi}_{G_1} \right) & 0 \otimes \left( \boldsymbol{\Phi}_{G_T} \boldsymbol{\Psi}_{G_2} \right) \ldots \end{bmatrix}
$$

$$
= \begin{bmatrix} \left( \boldsymbol{\Psi}_{G_1}^* \boldsymbol{\Psi}_{G_1} \right) & 0 \ldots \\ 0 & \left( \boldsymbol{\Psi}_{G_2}^* \boldsymbol{\Psi}_{G_2} \right) \ldots \\ \vdots & \\ 0 & 0 \ldots \end{bmatrix}
$$

Part 1: If $\boldsymbol{\Psi}_{G_1}$ is unitary then $\boldsymbol{\Psi}_{G_1}^* = \boldsymbol{\Psi}_{G_1}^{-1}$. Thus in this case $\boldsymbol{\Phi}_D \boldsymbol{\Psi}_D = I_{NT}$ which implies $\boldsymbol{\Phi}_D = \boldsymbol{\Psi}_D^* = \boldsymbol{\Psi}_D^{-1}$ implying $\boldsymbol{\Psi}_D$ is unitary.

Part 2: Considering $\boldsymbol{\Psi}_D$ is unitary whic means $\boldsymbol{\Phi}_D = \boldsymbol{\Psi}_D^* = \boldsymbol{\Psi}_D^{-1}$. Thus $\boldsymbol{\Phi}_D \boldsymbol{\Psi}_D = I_{NT}$ and so $\boldsymbol{\Psi}_{G_i}^* \boldsymbol{\Psi}_{G_i} = I_N \rightarrow \boldsymbol{\Psi}_{G_i}^{-1} = \boldsymbol{\Psi}_{G_i}^*$. $\therefore \boldsymbol{\Psi}_{G_i}$ is unitary proving the 2nd part and completing the proof. $\qquad \square$

**Property 8.** *EFT is invariant to the order of application of DFT or GFT on signal X.*

The above property can be observed from equation 6 using the fact that matrix multiplication is associative.

# D  DATASETS

Table 3: The statistics of the Large scale Dynamic graph datasets for link prediction.

| SR Datasets | Beauty | Games | CDs |
|---|---|---|---|
| # of Users | 52,024 | 31,013 | 17,052 |
| # of Items | 57,289 | 23,715 | 35,118 |
| # of Interactions | 394,908 | 287,107 | 472,265 |
| Average length | 7.6 | 9.3 | 27.6 |
| Density | 0.01% | 0.04% | 0.08% |

**Continuous Time Dynamic Graph link prediction dataset in sequential recommendation setting:** For showing the efficacy of our method on large dynamic graphs, we perform experiments on three real-world e-commerce datasets (cf., Table 3) for sequential recommendation. Specifically, we pose the sequential recommendation as a link prediction problem on temporal graphs. The penultimate and last interactions are used for validation and testing, respectively. The graphs at each interaction timestamp is constructed as detailed in (Zhang et al., 2022) i.e., at time $t$, the subgraph ($G_t$) containing all interactions till $t$ is considered. Then the $m$-hop neighborhood $G_t^m(u)$ around the user $u$ is sampled from it. The next item to predict is the item ($i_{t+1}$) interacted with at time $t + 1$. Thus the training set would contain $(G_1^m(u), i_2), (G_2^m(u), i_3) \ldots (G_{T-2}^m(u), i_{T-1})$ and the test set would have $(G_{T-1}^m(u), i_T)$. The graph construction is done in the preprocessing phase to speed-up training and testing.

Table 4: Statistics and details for link prediction on the benchmark dynamic graph datasets. LP is the abbreviation for Link Prediction and NC is for Node Classification.

| | # Nodes | # Edges | # Time Steps (Train / Val / Test) | Task |
|---|---|---|---|---|
| SBM | 1,000 | 4,870,863 | 35 / 5 / 10 | LP |
| UCI | 1,899 | 59,835 | 62 / 9 / 17 | LP |
| AS | 6,474 | 13,895 | 70 / 10 / 20 | LP |
| Elliptic | 203,769 | 234,355 | 31 / 5 / 13 | NC |
| Brain | 5,000 | 1,955,488 | 10 / 1 / 1 | NC |

**Benchmark Dynamic Graph Datasets**: Table 4 summarizes datasets for link prediction on benchmark dynamic graph datasets. Each dataset contains a sequence of time-ordered graphs. SBM is a synthetic dataset to simulate evolving community structures. UCI dataset is a student community network where nodes represent the students, and the edges represent the messages exchanged between them. AS dataset summarizes a temporal communication network indicating traffic flow between routers. The Elliptic (Ell) dataset delineates legitimate versus unlawful transactions within the elliptic network of Bitcoin transactions. In this context, nodes symbolize individual transactions, while edges correspond to the pathways of monetary transfers. The Brain (Brn) dataset focuses on nodes representing minuscule cerebral regions or cubes, with the edges signifying their interconnections.

**Synthetic Dataset** Consider the dynamic graph over T timesteps. Thus we have T graph snapshots. We compute the eigenvectors at each snapshot and place them over the graph's nodes. Moreover, for each node (spread over T timesteps) we compute a periodic signal that is added to the eigenvector component $Evec(G_t)$. So the expression for the noise added to the signal would be: $X(i, t) = \sum_k \alpha_k Evec(G_t)[i, k] + \sum_f \beta_f * e^{i\omega_t f}[t]$. In our experiments, we have used only one randomly chosen eigenvector. Also we consider only a single sinusoid frequency $\omega$. $\alpha_k, \beta_f$ are parameters and are set to $\frac{1}{2}$ in our experiments. For noise, we add to $X(i, t)$ a signal taken from a Gaussian distribution with 0 mean i.e. $X(i, t) = X(i, t) + \mathcal{N}(0, \delta)$, where $\delta$ is the standard deviation.

## D.1  EXPERIMENTAL SETUP

We implement our models using the DGL framework (Wang et al., 2019) in the pytorch library (Paszke et al., 2017). The hyperparameters are selected from the following search space: learning rate $\in [0.01, 0.0003]$, $l_2$ regularization parameter $\alpha \in [0.01, 0.00001]$, embedding and hidden layer dimensions $\in \{32, 64, 128\}$, filter order $\in \{2, 4, 8, 16\}$, subgraph size $\in \{1, 2, 3, 4\}$. The experiments are run on a single Tesla P100 GPU. We run our method for 5 runs per dataset and report the mean of the results. For the baselines we report the best results that have been reported unless mentioned otherwise. If results are not available we run baselines by using the implementation provided with default parameters and optimizing the hidden size (width) and layer number (depth) of the network. Regarding graph construction, for the Sequential Recommendation (SR) datasets we use similar to (Zhang et al., 2022). For the Session Based Recommendation (SBR) setting we use the transition graph of the items in the sequence as in (Wu et al., 2019). We also try with higher order graphs, albeit without any gains, as reported in (Wu et al., 2019). Moreover, since for SBR the last

item is of more significance to the prediction task and the datasets suffer from overfitting we modify the prediction layer accordingly and incorporate appropriate changes from baselines.

## D.2 IMPLEMENTATION

We intend to perform filtering in spectral space for dynamic graphs using *EFT*. Since our idea is to perform collective filtering along the vertex and temporal domain in *EFT*, we need two modules to compute $\Psi_{G_t}$ (vertex aspect) and $\Psi_T$ (temporal aspect), respectively, in equation 6 of *EFT*. We now explain these modules in detail.

**Filtering along the vertex domain:** This module computes the convolution matrix $\Psi_{G_t}$ in equation 6. Consider the filter response $\hat{\Lambda}_l$ which is a diagonal matrix with diagonal values representing the magnitude of the corresponding frequency(eigenvalue). In order to avoid the computational cost of the eigendecomposition, we choose to approximate the it using polynomials. In this work, we use the Chebyshev polynomials (Defferrard et al., 2016). Specifically, the frequency response of the desired filter is approximated as $\hat{\Lambda}_l = \sum_{k=0}^{O_f} c_k T_k(\tilde{\Lambda})$, where $O_f$ is the polynomial/filter order, $T_k$ is the Chebyshev polynomial basis, $\tilde{\Lambda} = \frac{2\Lambda}{\lambda_{max}} - I$, $\lambda_{max}$ is the maximum eigenvalue and $c_k$ is the corresponding *filter coefficients*. Thus, we can approximate the filtering operation as: $X * \Lambda_l \approx U \left( \sum_{k=0}^{O_f} c_k T_k(\tilde{\Lambda}_l) \right) U^* X = \sum_{k=0}^{O_f} c_k T_k(U\tilde{\Lambda}_l U^*)X = \sum_{k=0}^{O_f} c_k T_k(\tilde{L})X$. Having the filter coefficients $c_k$ as learnable parameters enables learning of filter for the task. The convolution $X * \Lambda_l$ gives the desired filtered response.

**Filtering along the temporal Domain:** After performing filtering in the vertex domain, we aim to filter over the temporal signals using $\Psi_T$ as in equation 6. To apply the $\Psi_T$ (Fourier transform), we must first ensure that the signals in sequences are sampled at uniform intervals. In the continuous time setting, interactions between nodes could occur at anytime or the sampling could be non-uniform, Thus, we perform a mapping from $R^{T \times d} \to R^{T \times d}$ that aims to map the input space to a uniformly sampled space. For computational reasons, we select the current and next embeddings (with positional information) along with the timestamp information ($E_t(t) \in R^d$) for getting the mapped embedding akin to interpolation. Formally, let $X_t^i \in R^d$ be the embeddings of the node at time $t$. This is first mapped to the interpolated space using a universal approximator: $X_t = W_2^i \sigma^i(W_1^i[X_t^i; X_{t+1}^i; E_t(t)] + b_1^i) + b_2^i$, where $W_1^i, W_2^i, b_1^i, b_1^i$ are learnable parameters and $\sigma^i$ is a non-linearity. We call this module the *time encoding layer*, which is essential for applying Fourier transform along the temporal dimension. Let $X = X_t \in R^{T \times d}$ be the interpolated sequence of embeddings of the node. This is converted to the frequency domain ($\hat{X} \in R^{T \times d}$) using the DFT matrix $\Psi_T$ as $\hat{X} = \Psi_T X_t$ Then we multiply $\hat{X}$ element-wise by a temporal filter $F_T \in R^{T \times d}$ to obtain the filtered signal $\hat{X}_f = F_T \odot \hat{X}$ which is then converted back to the temporal domain by using the inverse transform $\Psi_T^*$ to get $X_f = \Psi_T^* \hat{X}_f$. $X_f$ is the equivalent of $\hat{X}_G$ in equation 6 that is the output of *EFT*. In practice, the fast Fourier transform is used that can perform the computations in order $\mathcal{O}(T log(T))$. Hence, overall time complexity of the architecture is $O((N + E)T + NTlogT)$. To map the output back to the original space from the interpolated space we would need further mapping layers. Similar to (Zhou et al., 2022), we use the standard layer normalization (LN) and feedforward (FFN) layers: $X_F = \text{LN}\left(\text{LN}\left(X_t + \text{D}(X_f)\right) + \text{D}\left(\text{FFN}\left(\text{LN}\left(X_t + \text{D}(X_f)\right)\right)\right)\right)$, where $W_2^f, W_1^f, b_1^f, b_2^f$ are learnable parameters and D(.) represents dropout. We could stack filter layers with the node embeddings obtained from previous layers as inputs. $X_F$ is the final filtered signal that is used in the downstream prediction. For the concerned node $n$ we denote this as $X_F^n$.

## D.3 COMPUTATIONAL COMPLEXITY

Considering the spectral transform, the exact eigendecomposition of the joint laplacian would take order $\mathcal{O}((NT)^3)$ whereas our method of *EFT* would take $\mathcal{O}(N^3 T + NT \log(T))$. Thus we reduce the complexity from a factor of $T^3$ to $T \log(T)$. This would be beneficial in cases where there are many timesteps considered. For the model at the implementation level since we have made use of a function approximator that runs in time linear to the number of edges ($\varepsilon$), the time complexity is $\mathcal{O}(\varepsilon + NT \log(T))$. We have performed a wall clock run time analysis for the training of our method and the results in table 5 shows that it is comparable to a dynamic graph based baseline (that doesn't use any spectral transform):

Table 5: Wall clock running (sec/epoch) time of our and baseline method on SR datasets

| Dataset | Method | Wall clock time (sec/epoch) |
|---------|--------|------------------------------|
| Beauty | DGSR | 565 |
| Beauty | EFT | 753 |
| Games | DGSR | 1719 |
| Games | EFT | 2535 |
| CD | DGSR | 5415 |
| CD | EFT | 12637 |

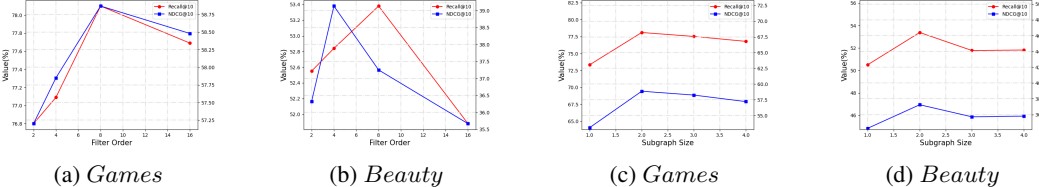

| (a) *Games* | (b) *Beauty* | (c) *Games* | (d) *Beauty* |
|---|---|---|---|

Figure 6: Effect of the parameters (filter order and subgraph size) on *EFT* performance.

# E ABLATION STUDY

Table 6: Ablation study of our model. We report Recall@10 (R@10) and NDCG@10 on Beauty and Games datasets.

|  | Beauty | | Games | |
|---|---|---|---|---|
|  | R@10 | NDCG@10 | R@10 | NDCG@10 |
| EFT | **53.23** | **37.10** | **77.78** | **58.75** |
| w/o Temporal filter | 52.42 | 36.12 | 76.55 | 56.95 |
| w/o Graph filter | 38.27 | 24.39 | 58.36 | 40.06 |
| +High Pass Filter | 47.53 | 31.10 | 76.88 | 57.24 |
| +Low Pass Filter | 52.71 | 36.76 | 77.74 | 58.49 |
| +Band Pass Filter | 52.27 | 36.09 | 76.67 | 56.98 |
| +Band Stop Filter | 45.34 | 29.09 | 77.63 | 58.42 |

**Component ablation:** In our first ablation study, we study the effect of various modules of the *EFT* architecture by systematically removing model components. Table 6 summarizes our findings. For example, performance declines when we remove the graph filtering module ("w/o graph filter" in Table 6). It confirms that the graph filters help to reduce long-range noise and positively impact performance. Next, we replace learnable filters of *EFT* with several static filters such as highpass, bandpass, lowpass, etc. Performance with static filters is less than that of dynamic filters, supporting our choice of having learnable filters in *EFT* .

**Parameter Selection:** In this experiment, we study the effect of filter and graph construction parameters that will help select optimal parameters for the model. Specifically, we run experiments for 1) the order of the graph filter and 2) subgraph size, which is the number of hops considered around the given user node for constructing the graph. The results are in Figure 7 with apt transform of the 2 metric scales for comprehension. For the filter order, we observe that for both datasets, there exists an optimal filter order at which the best performance is achieved. We observe that increasing filter order further causes overfitting on these datasets. For the subgraph size, we observe an increasing trend in the results, indicating that higher subgraph sizes ($> 1$) benefit the performance over a single hop (which is the sequence itself). This shows that modeling the SR as a graph learning problem is helpful over considering only the sequence. We conclude that beyond the subgraph size of two, the results saturate for these datasets.

