# OpenReview forum: "Beyond Spatio-Temporal Representations: Evolving Fourier Transform for Temporal Graphs"
_ICLR.cc/2024/Conference — ICLR 2024 poster_

### Official Review · Reviewer_qJBV · 2023-11-01

**Soundness:** 3 good
**Presentation:** 3 good
**Contribution:** 3 good
**Rating:** 6
**Confidence:** 3

**Summary:**

This paper proposes a new approach, EFT, to transform temporal graphs into the frequency domain, which can capture spectra of evolving graph structures. EFT provides solid theoretical proof of utilizing the Laplacian of the continuous time dynamic graph and pseudo-spectrum relaxations to decrease the computation cost. The downstream experiments on large-scale and standard dynamic graph datasets demonstrate the superior performance and efficiency of the proposed model.

**Strengths:**

1. This work proposed EFT to transform the evolving temporal graphs into the spectral domain, grounded on theoretical foundations. The EFT is very effective with the computational complexity of O(T+Tlog(T)).
2. The experiments in simulating scenarios clearly demonstrate that EFT is effective in filtering noise and amplifying useful signals in evolving temporal graphs.
3. EFT is of good interpretability; it can decompose the transform into the individual transforms of each domain to give clear relations from both the time and vertex domains.

**Weaknesses:**

The generalization of the proposed method needs to be clarified further. Even considering Theorem 1, the proposed method depends on the approximate \Psi_{AD}, the eigendecomposition of L_{J_d}. It is unclear whether any limitations associated with this will be introduced.

**Questions:**

1. The timespan of edges is a natural attribute of a temporal graph. Some recurrent works [1] [2] show that embedding the timespan of edges is important, especially in the sequential recommendation. I am wondering whether EFT could embed the timespan of edges and how. Such discussion may help the audience have better ideas on applying or extending EFT in solving their own problems in different application domains.
[1] Time-aware Dynamic Graph Embedding for Asynchronous Structural Evolution, TKDE'23.
[2] Time lag aware sequential recommendation, CIKM'22
2. In section 5.1, ”However, the same holds true for higher dimensions as well by conducting the EFT dimension-wise.” What does the dimension-wise mean?

---

> ### Author Response · Authors · 2023-11-16
> **Rebuttal by Authors**
>
> Dear Reviewer qJBV,
>
> We sincerely thank you for spending the time to review our work and for providing constructive reviews. We provide our answers point-wise (P1, P2...Pn) and seek guidance/advice on making suggested changes in the revised version.
>
> > P1:  the proposed method depends on the approximate \Psi_{AD}, the eigendecomposition of L_{J_d}. It is unclear whether any limitations associated with this will be introduced.
> -  One known limitation of eigendecomposition is that it is invariant to the sign/direction of the basis vector. This may cause issues with learning since the basis is not normalized. This can be resolved by the sign and basis invariant methods [1]. Moreover \Psi_{AD} is not designed to handle signed and directed graphs. Future works can explore how to generalize the laplacian as well as the proposed EFT to generic signed and directed graphs using techniques proposed in [2] and [3]. Do you recommend we add this discussion to the main paper?
>
> > P2: I am wondering whether EFT could embed the timespan of edges and how.
> -  The proposed neural architecture could be adapted to make use of timespan information as edge features for learning in the SR datasets. This could be an interesting future direction to explore the application of EFT along with task-specific features to enhance downstream performance. However here we focus on a joint spectral transform for dynamic graphs and to show its benefits on dynamic graphs, but we agree that the methods can further augment the method using task-specific features.
>
> > What does the dimension-wise mean?
> - Dimension-wise means the feature dimension of a node. Each node may have a multidimensional signal residing on it and the EFT can be independently applied to each channel or dimension for the dynamic graph. We shall add this in the revised version of the paper.
>
> We thank you again for spending time on our paper and providing reviews. Depending on your advice, we will add proposed changes to the next version of the paper. We look forward to your reply.
>
> Kind regards,
>
> Authors
>
>
> **References:**
>
> [1] https://arxiv.org/abs/2202.13013
>
> [2] https://arxiv.org/abs/2102.11391
>
> [3] https://proceedings.neurips.cc/paper/2016/file/7bc1ec1d9c3426357e69acd5bf320061-Paper.pdf

---

> ### Comment · Reviewer_qJBV · 2023-11-20
>
> I thank the authors for their detailed responses. The responses have addressed my concerns and comments. I would suggest the authors include those discussions in the main paper and give some tips to the followers about exploring the application of EFT along with task-specific features to enhance downstream performance. Good job, and I will remain my score.
>
> Reviewer qJBV

---

### Official Review · Reviewer_4jzq · 2023-11-01

**Soundness:** 4 excellent
**Presentation:** 3 good
**Contribution:** 3 good
**Rating:** 6
**Confidence:** 2

**Summary:**

This paper designed a new form of (approximated) Fourier Transform to analyze spatio-temporal graph signals with low computational complexity. The gist of the proposed approach lies in the decomposition of the spatial/graph domain and the temporal domain, reducing the computational complexity from $O((NT)^3)$ (performing GFT/SVD over the entire spatio-temporal domain) to $O(N^3T + NT \log(T))$. Furthermore, the proposed transform was shown to be robust to structural perturbations, and comprehensive simulations verified the utility of the transform when combined with some vanilla transformer models to perform link prediction tasks.

**Strengths:**

Just like in classical signal processing, it is not surprising that denoising and then reconstructing (predicting) the signal from the spectral domain can be much easier than dealing with the spatio-temporal signal directly. However, performing GFT over the entire spatio-temporal graph is typically computationally expensive, which hinders the spectrum analysis of spatio-temporal signals. This paper provided solid theoretical guarantees for using Evolving Graph Fourier Transform (EFT: DFT+GFT in order) to approximate the absolute decomposition (AD) with reasonably low approximation error. Standard properties of the transform are established in section 5.1 as well. For the simulations, it is also very interesting to see that EFT can be used within transformer models to perform sequential link prediction tasks in practice.

**Weaknesses:**

The idea of decoupling spatial and temporal domains in spatio-temporal data analysis is actually not new, especially for the spatio-temporal graph neural networks (ST-GCN) community. As a matter of fact, most existing ST-GCN models, no matter their design come from empirical perspectives [1, 2, 3] or theoretical perspectives [4, 5], all share the idea of graph-time decoupling to reduce the computational complexity. However, only a handful of prior works on ST-GCNs have been mentioned and discussed in the paper. I understand that the authors are considering graphs that change with time, for which most ST-GCNs are not directly applicable as they deal with static graphs, but I still believe related discussions should be added so that the readers are clearer about the contribution of this paper. Furthermore, the complexity of the proposed approach seems quite similar to those in [5]. The authors may also want to have some comments on that as well.

In addition, for the approximation of EFT to work, two extra assumptions are needed in place: 1) The rate of change of the graph with time is bounded; 2) The eigenvalues of the graph Laplacian at any given timestep and between timesteps have a multiplicity of 1. I think it is better to state these assumptions in the main text instead of the appendix to help the readers better understand the limit of the approach, as the assumptions are quite important (and intuitive) for the EFT to work.

[1] Yan, Sijie, Yuanjun Xiong, and Dahua Lin. "Spatial temporal graph convolutional networks for skeleton-based action recognition." Proceedings of the AAAI conference on artificial intelligence. Vol. 32. No. 1. 2018.

[2] Yu, Bing, Haoteng Yin, and Zhanxing Zhu. "Spatio-temporal graph convolutional networks: A deep learning framework for traffic forecasting." arXiv preprint arXiv:1709.04875 (2017).

[3] Cao, Defu, et al. "Spectral temporal graph neural network for multivariate time-series forecasting." Advances in neural information processing systems 33 (2020): 17766-17778.

[4] Pan, Chao, Siheng Chen, and Antonio Ortega. "Spatio-temporal graph scattering transform." arXiv preprint arXiv:2012.03363 (2020).

[5] Kartal, Bünyamin, Eray Özgünay, and Aykut Koç. "Joint time-vertex fractional fourier transform." arXiv preprint arXiv:2203.07655 (2022).

**Questions:**

1. In addition to the Fourier Transform, there appears to be research interests in performing spectral clustering for dynamic networks as well [6, 7]. Since node clustering can also be used to predict missing links, what advantages and disadvantages do you think EFT will have compared to spectral clustering?

2. Just curious, beyond link prediction, what changes would we need if we also want to predict node features?

[6] Liu, Fuchen, et al. "Global spectral clustering in dynamic networks." Proceedings of the National Academy of Sciences 115.5 (2018): 927-932.

[7] Martin, Lionel, Andreas Loukas, and Pierre Vandergheynst. "Fast approximate spectral clustering for dynamic networks." International Conference on Machine Learning. PMLR, 2018.

---

> ### Author Response · Authors · 2023-11-16
> **Rebuttal by Authors**
>
> Dear Reviewer 4jzq,
>
> We sincerely thank you for spending the time to review our work and for providing constructive reviews. We provide our answers point-wise (P1, P2...Pn) and seek guidance/advice on making suggested changes in the revised version.
>
> > P1: Regarding related works
> - We completely agree here and also share the same sentiments. We have incorporated and discussed the related works in the revised version of the paper.
>
> > P2:  I think it is better to state these assumptions in the main text
> - Thank you for your feedback on the presentation. We have incorporated the changes in the revised version of the paper.
>
> > P3: Regarding spectral clustering and advantages of EFT over it
> - We would like to remark that EFT is a transform (like GFT) and in order to use it in representation learning we would need to further process the transformed signals for example by filtering etc. Spectral clustering is an algorithm for clustering that uses the GFT transform for static graphs. While recognizing the inherent differences between EFT and spectral clustering, it seems challenging to draw direct comparisons, however, it is an important research direction raised by the reviewer to explore EFT for benefits in spectral clustering in the case of dynamic graphs. As in the standard method, we could compute the EFT of a dynamic graph, then use a clustering method in order to perform the clustering using the EFT rows as “feature vectors”. Thus EFT could be used to compute the spectral clustering. Also, future works could explore using the state-of-the-art method to approximately compute the GFT of graphs at future timesteps that can be used to compute the EFT which can further be used for fast spectral clustering and other applications.
>
> > Just curious, beyond link prediction, what changes would we need if we also want to predict node features?
> -  We have used EFT in our experiments to perform node classification. Specifically, two of the datasets (Elliptic, Brain) used in Table 2 are node classification datasets. To contrast with link prediction: In link prediction, we provide signals from the two participating nodes as input to the task-specific layer eg: concatenation of node features followed by MLP layer. For node classification, we only provide the node under consideration to the downstream task-specific layer which could be an MLP. To remark, the EFT transform can be used to learn generic spectral representations of dynamic graphs that can be applied in any downstream dynamic graph task.
>
> We thank you again for spending time on our paper and providing reviews. Depending on your advice, we will add proposed changes to the next version of the paper. We look forward to your reply.
>
> Kind regards,
>
> Authors

---

> > ### Comment · Reviewer_4jzq · 2023-11-22
> >
> > Thanks for the response from the authors. I do not have further questions and will maintain the score.

---

### Official Review · Reviewer_sEtv · 2023-11-04

**Soundness:** 3 good
**Presentation:** 1 poor
**Contribution:** 3 good
**Rating:** 6
**Confidence:** 4

**Summary:**

This paper proposes a method called evolving Fourier transform (EFT) for spectral analysis of dynamic graphs that is formulated as an optimization problem over the Laplacian of the continuous time dynamic graph. At implementation level it performs vertex domain filtering via Chebyshev polynomial approximation of graph filters followed by a classical time domain filtering via Fourier transform. Although the experiments are limited but the method is novel. The paper is dense and difficult to read at times. It is a novel work but could have been presented in a more elaborated way by providing more examples, intuitions, connections to FT and GFT via examples.

**Strengths:**

Although the experiments are limited, the method is novel.

**Weaknesses:**

The paper is dense and difficult to read at times. It is a novel work but could have been presented in a more elaborated way by providing more examples, intuitions, connections to FT and GFT via examples.

A table or example for clarifying the notations will significantly improve the readability. Please provide explanations/intuitions of all the terms in Equation (4) when the EFT is defined first.

The coupling of filtering in vertex and time domains is explained in Appendix C.2 which should be moved to the main text.

The paper does not state its limitations, for example non-applicability to dynamic graphs with node addition or node drops.

Graph signal variation equations can be displayed with number since it the the basic term used in further analysis.

**Questions:**

It would help the readers if more insights are given in terms of small toy examples. What will be harmonics for ring of ring graphs (no edge addition or dropping over time)? Does this have some relation to classical Fourier transform?

---

> ### Author Response · Authors · 2023-11-16
> **Rebuttal by Authors**
>
> Dear Reviewer sEtv,
>
> We sincerely thank you for spending the time to review our work and for providing constructive reviews. We provide our answers point-wise (P1, P2...Pn) and seek guidance/advice on making suggested changes in the revised version.
>
> > P1: Regarding presentation
> - Thank you for your feedback on the presentation. We have incorporated the changes in the revised version of the paper. Regarding the example for showing connection to dft and gft we think the example of ring of ring graphs perfectly is able to capture the phenomena. Do you suggest we incorporate it in the paper? Once you confirm, we can add that to the final camera-ready version of the paper as a running example.
>
> > P2: It would help the readers if more insights are given in terms of small toy examples. What will be harmonics for ring of ring graphs (no edge addition or dropping over time)? Does this have some relation to classical Fourier transform?
> - For a ring graph we can show that the laplacian is a circulant matrix and the eigendecomposition gives the basis vectors of the DFT(classical Fourier transform) matrix with the eigenvalues the corresponding frequencies. For a ring of ring graphs, we can show (using property 1) that the EFT matrix comes out to be the Kronecker product of two DFT matrices (one along the time domain and one along the vertex domain). Similarly, the harmonics will be the Kronecker sum of the harmonics along the individual domains. To remark, we can say that the GFT is a generalization of DFT and the EFT is a generalization of GFT, for dynamic graphs. We have added discussions of this in the revised version of the paper.
>
> We thank you again for spending time on our paper and providing reviews. Depending on your advice, we will add proposed changes to the camera-ready version of the paper. We look forward to your reply.
>
> Kind regards,
>
> Authors

---

> > ### Comment · Reviewer_sEtv · 2023-11-17
> > **Requesting more inputs from authors**
> >
> > Could you please respond to the weakness section?

---

> > > ### Author Response · Authors · 2023-11-17
> > > **Author Response**
> > >
> > > Dear Reviewer sEtv,
> > >
> > > Thank you for the discussion. We should have addressed the rebuttal to point P1 in detail. We have made the changes, suggested by you in the weakness section. Here we discuss in detail:
> > >
> > > > W1: The paper is dense and difficult to read at times. It is a novel work but could have been presented in a more elaborate way by providing more examples, intuitions, and connections to FT and GFT via examples.
> > > - Thank you for your feedback on the presentation. We have incorporated the changes in the revised version of the paper, in section 5 page 6. Regarding the example for showing connection to dft and gft we think the example of ring of ring graphs perfectly is able to capture the phenomena. Do you suggest we incorporate it in the paper? Upon your confirmation, we will add it to the final camera-ready version of the paper as a running example.
> > >
> > > > W2: A table or example for clarifying the notations will significantly improve the readability. Please provide explanations/intuitions of all the terms in Equation (4) when the EFT is defined first.
> > > - We appreciate your suggestion to include a table for clarity. Unfortunately, due to page constraints, we could not add a table. However, we address it by providing detailed notations in Section 3. If further clarification is required, we are open to suggestions.
> > >
> > > > W3: The coupling of filtering in vertex and time domains is explained in Appendix C.2 which should be moved to the main text.
> > > - We have incorporated the changes to the main paper (cf. sec 5.1).
> > >
> > > > W4: The paper does not state its limitations, for example, non-applicability to dynamic graphs with node addition or node drops.
> > >
> > > We address the limitation regarding the non-applicability of dynamic graphs with node addition or node drops in Section 5, page 5 of the theory section. Additionally, in discussions with reviewer qJBV, we recognize a known limitation related to eigendecomposition’s invariance to the sign/direction of the basis vector. We acknowledge the considerations from this invariance and suggest addressing it through sign and basis invariant methods [1]. In the current scope, we do not consider generic signed and directed graphs. To mitigate this, we suggest future works explore generalizing the Laplacian and the resulting transform to such graphs, leveraging techniques proposed in [2], [3], [4].
> > > Do you recommend we add this discussion to the main paper?
> > >
> > >
> > > > W5: Graph signal variation equations can be displayed with numbers since it the the basic term used in further analysis.
> > > -  We have added this recommended modification (equation 3, page 4).
> > >
> > > We are happy to discuss any further queries or concerns. We look forward to your reply.
> > >
> > > Kind regards,
> > >
> > > Authors
> > >
> > >
> > >
> > >
> > > **References**:
> > >
> > > [1] https://arxiv.org/abs/2202.13013
> > >
> > > [2] https://arxiv.org/abs/2102.11391
> > >
> > > [3] https://proceedings.neurips.cc/paper/2016/file/7bc1ec1d9c3426357e69acd5bf320061-Paper.pdf
> > >
> > > [4] https://www.jmlr.org/papers/volume22/20-1289/20-1289.pdf

---

### Official Review · Reviewer_YfD5 · 2023-11-05

**Soundness:** 3 good
**Presentation:** 3 good
**Contribution:** 3 good
**Rating:** 6
**Confidence:** 3

**Summary:**

This work proposes an invertible spectral transform that captures evolving representations on temporal graphs. EFT is built upon the approximation of the exact solution to the variational form. They provide theoretical bounds of the difference between EFT and the exact solution to the variational form. Their experiments show that EFT effectively filter out the noise signals and enhance task performance against the baselines.

**Strengths:**

The strengths of this work are summarized as follows:

(1) The idea of evolving graph Fourier transform is interesting, which provides an alternative way of extracting the information within spatial temporal graphs.

(2) The theoretical properties are analyzed with provable guarantees on the approximation properties of EFT.

(3) The benchmarks comparing this method with the baselines look promising.

**Weaknesses:**

The weakness of this work are summarized as follows:

(1) Despite giving guarantees on the approximation part, this work does not address the theoretical guarantees of learning under this module. Hence this makes this work more like heuristic. The reviewer believes that this might be quite difficult though.

(2) The font size of the tables should be improved. This makes it hard to read. The reviewer suggests reducing presentation on the properties of EFT in proposition 5.1 to make up for the space and delay the properties of this transform to the appendix instead.

**Questions:**

The question is how is the complexity of computing this transform. The reviewer did not see how fast it can be computed compared with other baselines and would like to see if it can be efficiently computed empirically.

---

> ### Author Response · Authors · 2023-11-16
> **Rebuttal by Authors**
>
> Dear Reviewer YfD5,
>
> We sincerely thank you for spending the time to review our work and for providing constructive reviews. We provide our answers point-wise (P1, P2...Pn) and seek guidance/advice on making suggested changes in the revised version. Meanwhile, we uploaded a version where new changes are marked with green text color.
>
> > P1: Despite giving guarantees on the approximation part, this work does not address the theoretical guarantees of learning under this module. Hence this makes this work more like heuristic. The reviewer believes that this might be quite difficult though.
> - The architecture is designed from the theoretical motivation. From the theory, we see how to decompose the joint Fourier Transform for the Dynamic graph into individual components of Fourier Transforms along the temporal aspect and vertex domain. The empirical approximation of the model further depends on the errors introduced by the components used to approximate the filters such as from the truncated Chebyshev polynomials etc. While we have shown from the empirical results of the neural architecture that learning the joint frequency components as proposed by the theory benefits the dynamic graph representation learning, it is an interesting point mentioned by the reviewer regarding deriving further bounds for the errors introduced by the approximation of the model implementation.
>
> We believe that, taking inspiration from previous works [1] we could work on further bounding the approximation ability of the model. Would you suggest we discuss this in the paper, especially as future work in the conclusion section?
>
>
> > P2: The font size of the tables should be improved.
> - Thank you for your feedback on the presentation. We have incorporated the changes in the revised version of the paper. The new changes are marked with the green text color.
>
> > P3: The question is how is the complexity of computing this transform. How fast it can be computed compared with other baselines and can it be efficiently computed empirically.
> - The computational complexity of the transform is $\mathcal{O}(N^3T + NTlog(T))$. However this reduces in the neural architecture where we use polynomial approximations. The overall computational complexity of the neural architecture is $\mathcal{O}(E + NTlogT)$.
> As suggested by the reviewer we also compute the wall clock running time of our method and compare it against baselines. The below table shows the time taken in seconds per epoch of our method (EFT) vs the average over the baselines. Our work significantly reduces the complexity at empirical level if simple eigenvector decomposition would have been considered against the proposed computationally efficient EFT.
>
> Dataset  | Average Baseline  | Full EVD | *EFT (our implementation)* |
> | :------------- | :----------: | -----------: | -----------: |
> |SBM| 68.35 | 1419.40  | 68.97 |
> | UCI | 275.24 |  2493.20  | 276.36 |
> | AS | 115.34 |   2614.59   | 116.60 |
> | Brain | 2.86 | 124.67  | 2.95  |
> | Elliptic | 70.12 | 14329.54  | 71.66  |
>
> From the above table we see that the running time of our method is comparable to the baselines and that empirically it can be computed efficiently. Note our implementation saves time over the full EVD version (which has a complexity cubic in the number of nodes).
>
> We thank you again for spending time on our paper and providing reviews. Depending on your advice, we will add proposed changes to the next version of the paper. We look forward to your reply.
>
> Kind regards,
>
> Authors
>
> **References:**
>
> [1] https://arxiv.org/abs/0912.3848

---

### Author Response · Authors · 2023-11-19
**General Response**

We thank the reviewers for the feedback and useful comments. In addition to detailed responses, in this general comment, we would like to summarise identified strengths and changes in the new revision addressing common weaknesses. The new additions in the manuscript are marked in green.

Strengths:

- **Method novelty and effectiveness**: "The idea of evolving graph Fourier transform is interesting,...within spatial-temporal graphs" (Reviewer YfD5). "The method is novel" (Reviewer sEtv). "EFT is effective in filtering noise and amplifying useful signals in evolving temporal graphs" (Reviewer qJBV).

- **Theoretical Analysis**: "The theoretical properties are analyzed with provable guarantees on the approximation properties of EFT." (Reviewer YfD5). "This paper provided solid theoretical guarantees for using Evolving Graph Fourier Transform" (Reviewer 4jzq).

- **Experimental results**: "The benchmarks comparing this method with the baselines look promising." (Reviewer YfD5).  "it is also very interesting to see that EFT can be used within transformer models to perform sequential link prediction tasks in practice."(Reviewer 4jzq)



In the updated manuscript, we made the following changes and uploaded a new version of the paper (changes are marked in green):

- **Section 2 (Related work)**: Discussed and incorporated related works based on the reviews  (Reviewer 4jzq).

- **Section 3 (Preliminaries)**: Added a paragraph explaining common notations to help with the equations in the theory (Reviewer sEtv)

- **Section 5 (Constructing an evolving graph Fourier transform)**: Described examples connecting DFT, GFT, EFT etc. Explained the coupling of filtering in vertex and time domains in **section 5.1** and moved the properties section to the **appendix** (Reviewer YfD5). Addressed the concern of the method for variable nodes and ways to handle them (Reviewer sEtv). Stated assumptions and conditions in the main paper, that were previously mentioned in the appendix (Reviewer 4jzq )


- **Table 1 and Table 2** Incorporated presentation changes to increase the font size of tables (Reviewer YfD5).

- **Section 8 (Conclusion)**: Additionally we have added the the limitations in the conclusion section as a direction for future work. (Reviewer sEtv, Reviewer qJBV)

- **Appendix**: Addressed the missing definitions ( Reviewer qJBV)

---

### Meta-Review · Area_Chair_kQwL · 2023-12-11

**Metareview:**

The paper proposes an algorithm for adapting Fourier Transforms for temporal graphs that evolve through time. The paper offers good theoretical and empirical novelties with provable guarantees, which are recognized by all reviewers. Beyond request to clarify the manuscript, there were questions by the reviewers regarding the learning part of the paper and whether the guarantees still apply, or discussion of limitation. That said, all reviewers recognize the worth of the paper, and thus I suggest to be accepted.

**Justification For Why Not Higher Score:**

See above.

**Justification For Why Not Lower Score:**

See above.

---

### Decision · Program_Chairs · 2024-01-16

Accept (poster)